# On the role of asymmetric molecular geometry in high-performance organic solar cells

Jinfeng Huang[1,2,10], Tianyi Chen[1,10], Le Mei[3,4,10], Mengting Wang[1,10], Yuxuan Zhu[5], Jiting Cui[6], Yanni Ouyang[7], Youwen Pan[1], Zhaozhao Bi [8], Wei Ma[8], Zaifei Ma [5], Haiming Zhu [6], Chunfeng Zhang [7], Xian-Kai Chen [4,9] ✉, Hongzheng Chen [1] ✉ & Lijian Zuo [1,2] ✉

Although asymmetric molecular design has been widely demonstrated effective for organic photovoltaics (OPVs), the correlation between asymmetric molecular geometry and their optoelectronic properties is still unclear. To access this issue, we have designed and synthesized several symmetric-asymmetric non-fullerene acceptors (NFAs) pairs with identical physical and optoelectronic properties. Interestingly, we found that the asymmetric NFAs universally exhibited increased open-circuit voltage compared to their symmetric counterparts, due to the reduced non-radiative charge recombination. From our molecular-dynamic simulations, the asymmetric NFA naturally exhibits more diverse molecular interaction patterns at the donor (D):acceptor (A) interface as compared to the symmetric ones, as well as higher D:A interfacial charge-transfer state energy. Moreover, it is observed that the asymmetric structure can effectively suppress triplet state formation. These advantages enable a best efficiency of 18.80%, which is one of the champion results among binary OPVs. Therefore, this work unambiguously demonstrates the unique advantage of asymmetric molecular geometry, unveils the underlying mechanism, and highlights the manipulation of D:A interface as an important consideration for future molecular design.

Organic photovoltaics (OPVs) show promise as the next-generation clean energy source, due to their merits of low-cost solution processability, superior flexibility, good selective absorption, high power-to-weight ratio, vivid colors, etc[1–3]. These allow OPV with more diverse applications to harmoniously merge with our daily life, e.g. building or vehicle-integration, portable or wearable energy source, etc[4–6]. Encouragingly, the device performance of OPV surged in the recent years, and currently, the best-certified device efficiencies are ~19.4%

[1]State Key Laboratory of Silicon and Advanced Semiconductor Materials, Department of Polymer Science and Engineering, Zhejiang University, Hangzhou 310027, PR China. [2]Zhejiang University-Hangzhou Global Scientific and Technological Innovation Center, Hangzhou 310014, PR China. [3]Department of Chemistry, City University of Hong Kong, Kowloon 999077, Hong Kong. [4]Institute of Functional Nano & Soft Materials (FUNSOM), Soochow University, Suzhou 215123 Jiangsu, PR China. [5]State Key Laboratory for Modification of Chemical Fibers and Polymer Materials, Center for Advanced Low-dimension Materials, College of Materials Science and Engineering, Donghua University, Shanghai 201620, China. [6]State Key Laboratory of Modern Optical Instrumentation, Key Laboratory of Excited-State Materials of Zhejiang Province, Department of Chemistry, Zhejiang University, Hangzhou, Zhejiang 310027, China. [7]National Laboratory of Solid State Microstructures, School of Physics, and Collaborative Innovation Center for Advanced Microstructures, Nanjing University, Nanjing 210093, China. [8]State Key Laboratory for Mechanical Behavior of Materials, Xi'an Jiaotong University, Xi'an Jiaotong University, Xi'an 710049, PR China. [9]Jiangsu Key Laboratory of Advanced Negative Carbon Technologies, Soochow University, Suzhou 215123 Jiangsu, PR China. [10]These authors contributed equally: Jinfeng Huang, Tianyi Chen, Le Mei, Mengting Wang. ✉e-mail: xkchen@suda.edu.cn; hzchen@zju.edu.cn; zjuzlj@zju.edu.cn

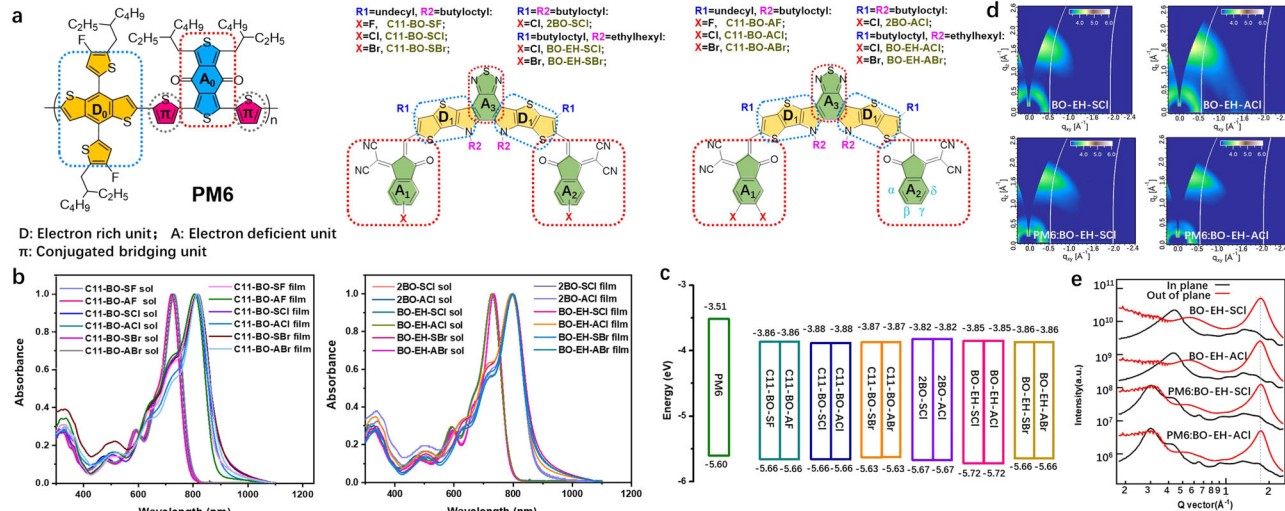

**Fig. 1 | Basic characteristics of the donor PM6 and six pairs of symmetric and asymmetric acceptors. a** Molecular structures of polymer donor PM6 (poly((4,8-bis(5-(2-ethylhexyl)-4-fluoro-thienyl)benzo[1,2-b:4,5-b′]dithiophene-2,6-diyl)-2,5-thiophenediyl(5,7-bis(2-ethylhexyl)-4,8-dioxo-4H,8H-benzo[1,2-c:4,5-c′]dithiophene-1,3-diyl)-2,5-thiophenediyl)) and acceptors (D represents electron-rich unit in blue dashed box, and A represents deficient unit in red dashed box and π represents conjugated bridging unit in the gray circle). **b** Normalized absorption spectra of twelve acceptors in CHCl₃ solution and thin films. **c** Energy level alignment of PM6 and all acceptors via CV measurement. **d** 2D GIWAXS images of pristine acceptor films and blend films. **e** GIWAXS intensity profiles of the corresponding films along the in-plane and out-of-plane directions.

and ~20.2% for single junction and tandem devices, respectively[7–9]. Nevertheless, the device performance of OPVs is still lagging behind that of their inorganic counterparts. It has been generally recognized that developing novel materials is the key to boosting the device performance of OPV.

The recent progress in OPV has witnessed the molecular evolution from fullerenes to Y6 derivatives, and this enables a dramatic increase in device efficiency[10]. Previous work has unveiled that the significant improvement in device performance with Y6 derivatives lies in their formation of delocalized excitons, longer exciton diffusion length, reduced non-radiative decay, etc., which are intimately related to their unique molecular packing behaviors and fine-tuned morphology and energetics. Besides, it is interesting to observe that from the fullerene family to the ITIC family and the Y6 family, the degree of molecular symmetry gradually reduces, from the ball-shaped (icosahedral) fullerenes, to ITIC molecules with a largely coplanar core and inversion symmetry, to Y6 molecules with a banana-shape and a helical chiral structure[7,11].

Following the above logic, the asymmetric molecular design with further reduction in symmetry seems to be a more advantageous structure, and has been intensively studied[11–16]. For example, Yan's group developed the asymmetric acceptor of BTP-2F-ThCl, which exhibits fine-tuned energy levels along with enlarged optical band gap relative to the symmetric acceptor BTP-4F. As a result, a higher open-circuit voltage ($V_{oc}$) and power conversion efficiency (PCE) are achieved[17]. Apart from tuning the photoelectric properties, the asymmetric acceptors also show obvious advantages in altering the morphology compared to the symmetric ones. For example, Kim et al. designed asymmetric acceptor IPC-BEH-IC2F, which shows excellent molecular packing and improved morphology characteristics, and enhanced photovoltaic properties[14]. Nevertheless, on the one hand, the current research typically takes the asymmetry as a chemical-tailoring tool to tune the energy levels, molecular packing, and morphology for high device performance. However, these effects can also be realized by using other strategies, such as modifying the side chains, backbones, etc. These smear the uniqueness of asymmetric molecular design, and leave the necessity of asymmetric molecular design questionable. On the other hand, the design of asymmetric molecular

structure is typically accompanied with the variation in energetic structures, molecular packing, morphology, etc., which poses a significant challenge to independently study the role of molecular asymmetry on the device performance of OPVs.

To exclusively examine the role of molecular asymmetry on carrier dynamics and device performance of OPVs, we have designed and synthetized six pairs of symmetric-asymmetric non-fullerene acceptors (NFAs), i.e., the C11-BO-SX and C11-BO-AX (X = F, Cl, Br), 2BO-SCl and 2BO-ACl as well as BO-EH-SX and BO-EH-AX (X=Cl, Br) (shown in Fig. 1a), which exhibit identical optical and electronic properties as well as film morphology, as confirmed by our experimental results. It is interesting to observe that the asymmetric NFA universally exhibits improved high $V_{oc}$ due to the suppressed non-radiative charge recombination, as verified by a variety of carrier dynamic measurements. Further, with molecular-dynamic (MD) simulation, we unveil that the asymmetric NFAs naturally exhibit more diverse molecular interaction structures at the D:A interface, as well as a higher charge-transfer (CT) energy. It is interesting to observe that the D:A interfacial structure with high CT energy takes the dominance. Moreover, the asymmetric NFAs exhibit less triplet state formation. These effects contribute to higher CT energy, suppressed non-radiative energy loss, and a high efficiency of 18.80%, which is one of the best among binary OPVs as shown in Supplementary Table 1. Our work highlights the uniqueness of molecular asymmetry in lowering non-radiative energy loss, which provides an instructive guideline for designing novel NFAs toward high device performance.

## Results
### Density functional theory (DFT) calculation, synthesis, and characterization of symmetric-asymmetric NFA pairs

We screened from a large pool of semiconductors to find out the appropriate asymmetric and symmetric isomeric pairs with similar energy levels, and finally identified the β or γ halogen position of end groups (2-(3-oxo-2,3-dihydro-1H-inden-1-ylidene) malononitrile, coded as IC) of the Y6 derivatives as our design principle (Fig. 1a). The symmetric NFAs feature one halogen atom on β or γ position of IC end groups, while the asymmetric NFAs deliver two halogen atoms on β and γ position of one IC terminal group, while the other end group is

halogen-free. Finally, we filtered six pairs of isomers as target NFAs and they all exhibited identical energy levels via DFT calculation (summarized in Supplementary Table 2), which is in accordance with cyclic voltammetry (CV) and ultraviolet–visible (UV–vis) spectra below. Different halogen positions (including α, β, γ, δ positions) of end groups usually have a varied impact on the energy levels of NFAs[18,19]. From the DFT calculations at B3LYP/6-31G(d) level (Supplementary Fig. 1), two simplified isomeric acceptors with chlorinated end groups at the β and γ positions show practically equal energy levels, but the α and δ positions of chlorinated end groups showed obviously higher the highest occupied molecular orbital (HOMO) and lowest unoccupied molecular orbital (LUMO) levels. The previous work also experimentally unveiled that the α position of chlorinated end groups showed significantly upshifted energy levels and enlarged energy band gaps[18]. Hence, we selected the NFAs pairs with β and γ positions of halogenated end groups. Since the length of alkyl chains has almost no effect on energy levels, we picked three sets of long and classic alkyl chains as side chains as shown in Fig. 1a. To further confirm the DFT results, we performed geometry optimizations calculations at the long-range corrected ωB97XD/6-31G (d, p) level (Supplementary Fig. 2). These pairs of Y-shape acceptors with simplified side chains as methyl groups also showed close HOMO and LUMO energy levels, which is beneficial for independently investigating the role of molecular asymmetry on the device performance of OPVs.

According to the DFT calculation, three mono-halogenated end groups (including 2-(5 or 6-fluoro-3-oxo-2,3-dihydro-1H-inden-1-ylidene)malononitrile, 2-(5 or 6-chloro-3-oxo-2,3-dihydro-1H-inden-1-ylidene) malononitrile and 2-(5 or 6-bromo-3-oxo-2,3-dihydro-1H-inden-1-ylidene)malononitrile) were adopted to make the symmetric acceptors (namely C11-BO-SX (X = F, Cl, Br), 2BO-SCl and BO-EH-SX (X = Cl, Br)) via Knoevenagel condensation reaction as shown in Supplementary Fig. 3. Meanwhile, non-halogenated 2-(3-oxo-2,3-dihydro-1H-inden-1-ylidene)malononitrile and double-halogenated end groups (including 2-(5, 6-fluoro-3-oxo-2,3-dihydro-1H-inden-1-ylidene)malononitrile, 2-(5, 6-chloro-3-oxo-2,3-dihydro-1H-inden-1-ylidene)malononitrile and 2-(5,6-bromo-3-oxo-2,3- dihydro-1H-inden-1-ylidene)malononitrile) were adopted to fabricate asymmetric acceptors (namely C11-BO-AX (X = F, Cl, Br), 2BO-ACl and BO-EH-AX (X = Cl, Br)) via the mole ratio control. The structures of all the acceptors were determined through nuclear magnetic resonance (NMR) spectrum and matrix-assisted laser desorption and ionization time of flight (MALDI-TOF) mass spectra (Supplementary Figs. 4–27).

The optical properties of all acceptors were expressed in absorption spectra as shown in Fig. 1b, Supplementary Fig. 28, and the related data were outlined in Supplementary Table 2. The acceptors showed maximum peaks at 722-731 nm in diluted CHCl$_3$ solutions. Every pair of symmetric-asymmetric NFAs showed the identical absorption region with maximum peaks as well as onsets. For example, BO-EH-SCl and BO-EH-ACl showed the same peaks and onsets at 727 and 784 nm, respectively, in a dilute solution. When converted into the solid state, apparent redshifts to 793-818 nm were observed, suggesting strong molecular aggregation. Every pair of acceptors as films showed a close onset wavelength ($\lambda_{onset}$) and an almost identical optical band gap. As for C11-BO-SX/C11-BO-AX (X = F, Cl, Br), the optical band gaps gradually decreased from F, Cl to Br (1.42 eV, 1.38 eV to 1.37 eV in films) based pairs. In the case of chloride acceptors in film, 2BO-SCl/2BO-ACl pair achieved the largest optical band gaps (1.43 eV) than C11-BO-SCl/C11-BO-ACl (1.38 eV) and BO-EH-SCl/BO-EH-ACl (1.41 eV) pairs. To verify the HOMO and LUMO energy levels of asymmetric and symmetric isomers pairs, the CV measurements were carried out and the corresponding energy levels alignment were depicted in Fig. 1c[20]. According to the CV curves in Supplementary Fig. 29, symmetric acceptors C11-BO-SX (X = F, Cl, Br), 2BO-SCl and BO-EH-SX (X = Cl, Br) exhibited almost the same HOMO and LUMO levels as asymmetric acceptors C11-BO-AX (X = F, Cl, Br), 2BO-ACl and BO-EH-

AX (X = Cl, Br), respectively. Compared to the six acceptors C11-BO-SX and C11-BO-AX (X = F, Cl, Br) with straight outer side chains, the acceptors 2BO-SCl, 2BO-ACl, BO-EH-SX and BO-EH-AX(X= Cl, Br) with branched side chains showed slightly upshifted LUMO levels, which is beneficial for realizing higher photovoltage in devices. The NFAs with a similar conjugated backbone showed different energy levels may result from diverse π–π stacking interactions via distinct side chains, which can induce dissimilar electronic coupling and energy gap[21]. Meanwhile, these NFAs with similar backbones but dissimilar side chains will display unequal morphologies, which might be profitable to generate different charge dynamics characteristics and thus PCEs[20].

## Film morphology and miscibility

Since the morphology has been demonstrated to have a great impact on the carrier dynamics and the device performance of OPVs[10], the morphology of the six symmetric-asymmetric pairs is studied to examine the possible effect of molecular symmetries on the molecular packing and phase structures of active layers. From the atomic force microscopy (AFM) measurement as shown in Supplementary Fig. 30, we observed the phase images of all the acceptors delivered decent phase-separation domains and pronounced interpenetrating network. While little difference can be detected between the symmetric and asymmetric acceptors, which show close root-mean-square (RMS) roughness as summarized in Supplementary Table 3. Further, we explored the molecular stacking, and measured the grazing-incidence wide-angle X-ray scattering (GIWAXS) pattern of the C11-BO-SBr/C11-BO-ABr and BO-EH-SCl/BO-EH-ACl symmetric-asymmetric pairs. As shown in Supplementary Fig. 31, the C11-BO-SBr and C11-BO-ABr-based pure or blend films showed the nearly-identical signal peak locations in both out-of-plane (OOP) and in-plane (IP) direction, indicating similar packing distance. The pristine acceptor films of BO-EH-SCl and BO-EH-ACl also showed the identical packing space (Fig. 1d, e). Furthermore, the GIWAXS patterns of PM6:BO-EH-SCl (CCL: 23.25 nm) and PM6:BO-EH-ACl (CCL:24.06 nm) blend films also showed little differences with crystal coherence length (CCL), and both blends adopted the face-on molecular orientations. These results verify the molecular symmetric-asymmetric structure can affect little on the morphology in our designed case.

Further, we performed contact angle measurements to study the surface tension of the donor and acceptor materials, as well as the calculated Flory–Huggins interaction parameters $\chi^{D-A}$s, following the equation of $\chi^{D-A} = \sqrt{\gamma_D} - \sqrt{\gamma_A}$, as shown in Supplementary Fig. 32 and summarized in Supplementary Table 3. The smaller $\chi^{D-A}$ indicated the stronger miscibility of the acceptor with polymer donor PM6. The $\chi^{D-A}$ values of symmetric SMAs showed negligible differences under 0.1 with the counterpart of the asymmetric ones, indicating similar miscibility of symmetric and asymmetric acceptors with donor PM6 and this agreed with the AFM results. Despite a little difference, no variation trends between symmetric and asymmetric NFAs were found, which may attribute to the inevitable testing errors. Overall, six pairs of symmetric-asymmetric acceptors showed similar absorption, energy levels, and film morphology, which is conducive to independently studying the effect of molecular geometry on the performance of OPV devices.

## Photovoltaic properties and charge generation

To study the effect of molecular symmetries on the photovoltaic properties of OPV, a conventional structure of ITO/PEDOT:PSS/PM6:acceptor/PDINN/Ag was designed. The specific preparation process and characterizations of devices are provided in the Methods part. The current density–voltage (J–V) curves of OPVs based on PM6:acceptor blends are displayed in Fig. 2a and the corresponding photovoltaic parameters are summarized in Table 1. Besides, Source data are provided as a Source Data file. There is no obvious variation in the fill factor (FF, within 2.5%) between symmetric and asymmetric acceptors.

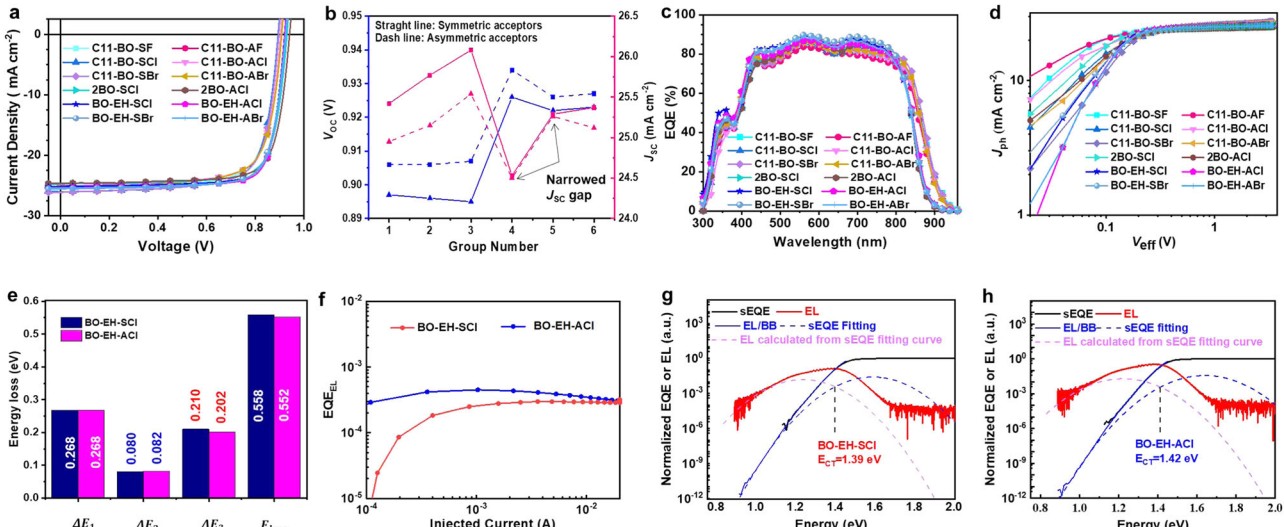

**Fig. 2 | Device performances and representative energy loss analysis. a** $J - V$ curves of OPVs under optimal conditions. **b** $V_{oc}$ and $J_{sc}$ values of different groups of isomers (Group No 1: C11-BO-SF and C11-BO-AF, Group No 2: C11-BO-SCl and C11-BO-ACl, Group No 3: C11-BO-SBr and C11-BO-ABr, Group No 4: 2BO-SCl and 2BO-ACl, Group No 5: BO-EH-SCl and BO-EH-ACl, and Group No 6: BO-EH-SBr and BO-EH-ABr). **c** EQE curves. **d** $J_{ph}-V_{eff}$ curves. **e** schematic diagram for energy losses and **f** $EQE_{EL}$ curves of BO-EH-SCl and BO-EH-ACl-based OPVs. sEQE and EL spectra of the **g** BO-EH-SCl and **h** BO-EH-ACl-based devices.

**Table 1 | Photovoltaic parameters of the OPVs devices based on different binary blends**

| Active layer | $V_{oc}$ [V] | $J_{sc}$ [mA cm$^{-2}$] | $J_{cal.}$ [mA cm$^{-2}$][a] | FF [%] | PCE [%][b] | $EQE_{EL}$ [%] |
|---|---|---|---|---|---|---|
| PM6:C11-BO-SF | 0.897 ± 0.002 | 25.42 ± 0.17 | 24.90 | 74.41 ± 0.36 | 16.96 ± 0.07 | 8.098 × 10$^{-4}$ |
| PM6:C11-BO-AF | 0.906 ± 0.001 | 24.95 ± 0.11 | 24.30 | 73.79 ± 0.66 | 16.68 ± 0.17 | 9.909 × 10$^{-4}$ |
| PM6:C11-BO-SCl | 0.896 ± 0.002 | 25.77 ± 0.21 | 25.25 | 73.99 ± 0.86 | 17.16 ± 0.18 | 5.542 × 10$^{-4}$ |
| PM6:C11-BO-ACl | 0.906 ± 0.002 | 25.15 ± 0.18 | 24.95 | 73.36 ± 0.68 | 16.79 ± 0.17 | 6.207 × 10$^{-4}$ |
| PM6:C11-BO-SBr | 0.895 ± 0.002 | 26.08 ± 0.10 | 25.59 | 73.21 ± 0.79 | 17.10 ± 0.17 | 7.537 × 10$^{-4}$ |
| PM6:C11-BO-ABr | 0.907 ± 0.001 | 25.54 ± 0.09 | 25.20 | 72.68 ± 0.32 | 16.84 ± 0.05 | 8.642 × 10$^{-4}$ |
| PM6:2BO-SCl | 0.926 ± 0.002 | 24.53 ± 0.14 | 24.45 | 77.87 ± 0.40 | 17.70 ± 0.16 | 2.363 × 10$^{-4}$ |
| PM6:2BO-ACl | 0.934 ± 0.003 | 24.50 ± 0.14 | 24.23 | 77.86 ± 0.30 | 17.83 ± 0.15 | 5.407 × 10$^{-4}$ |
| PM6:BO-EH-SCl | 0.922 ± 0.001 | 25.29 ± 0.10 | 25.07 | 78.99 ± 0.50 | 18.38 ± 0.13 | 2.902 × 10$^{-4}$ |
| PM6:BO-EH-ACl | 0.926 ± 0.001 | 25.26 ± 0.13 | 24.72 | 79.85 ± 0.22 | 18.68 ± 0.12 | 3.912 × 10$^{-4}$ |
| PM6:BO-EH-SBr | 0.923 ± 0.001 | 25.37 ± 0.13 | 25.28 | 76.95 ± 0.48 | 17.99 ± 0.17 | 5.491 × 10$^{-4}$ |
| PM6:BO-EH-ABr | 0.927 ± 0.002 | 25.12 ± 0.21 | 25.00 | 77.89 ± 0.17 | 18.11 ± 0.16 | 5.794 × 10$^{-4}$ |

[a]Integrated current densities from corresponding EQE curves.

[b]Average PCEs from over 10 devices.

Particularly, the symmetric-asymmetric pairs exhibit no unified variation trend between asymmetric and symmetric molecular structures. For example, the asymmetric acceptors-based OPVs displayed slightly smaller FF values than the symmetric counterparts in four symmetric-asymmetric pairs, i.e. the C11-BO-SX/ C11-BO-AX (X = F, Cl, Br) and 2BO-SCl/2BO-ACl, while the asymmetric acceptors BO-EH-SCl and BO-EH-SBr exhibited slightly larger FF values than symmetric acceptors (BO-EH-ACl and BO-EH-ABr). Interestingly, a very reliable variation trend on the short circuit current density ($J_{sc}$) and $V_{oc}$ values between symmetric and asymmetric pairs can be derived from both the average and champion results (Fig. 2b). It's found that all symmetric acceptors-based devices realized higher $J_{sc}$ than that of the asymmetric acceptors. To figure out the origin, we measured the external quantum efficiency (EQE). As shown in Fig. 2c, the integrated values ($J_{sc}$) from the EQE spectra match well with those from the $J-V$ curves, confirming that the molecular asymmetry tends to slightly reduce charge generation (Supplementary Fig. 33). These results indicate more efficient photons to free charge carrier generation. The charge generation properties were further investigated by measuring their photocurrent density ($J_{ph}$) versus effective voltage ($V_{eff}$) as shown in Fig. 2d and

Supplementary Table 4. Additionally, Source data are provided as a Source Data file. Here the saturated current density ($J_{sat}$) refers to the $J_{ph}$ when $V_{eff} = 3.0$ V, and all the symmetric acceptors delivered higher $J_{sat}$ than the asymmetric acceptors, respectively, indicating better charge generation properties of symmetric acceptors than those of asymmetric acceptors.

Moreover, even though six pairs of isomers have approximate energy levels, all the asymmetric acceptors (C11-BO-AX (X = F, Cl, Br), 2BO-ACl and BO-EH-AX (X= Cl, Br)) based devices universally delivered higher $V_{oc}$ (ca., 10 mV) than the counterpart of symmetric acceptors-base devices, as shown in Fig. 2b. Considering their identical band gap, the improved $V_{oc}$ is attributed to the suppressed charge recombination, which can be studied in detail in the next section. Above all, due to the improved balance between charge generation and recombination, the efficiency of binary OPV device performance shows an obvious difference between the symmetric and asymmetric one. As shown in Table 1, it is interesting to observe that the symmetric NFA-based devices show a higher PCE than the asymmetric with the undecyl side chains. While the asymmetric NFA-based devices show priority in PCE compared to that of the symmetric NFA with all branched side

**Table 2 | Detailed voltage loss ($V_{oc}$ loss) and related energy calculation for two representative systems**

| Active layer | $E_g^{opt}$ [eV] | $V_{oc}^{SQ}$ [V] | $V_{oc}^{rad}$ [V] | $V_{oc}$ [V] | $\Delta E$ [eV] | $\Delta E_2$ [eV] | $\Delta E_3$ [eV] | $E_{loss}$ [eV] | $E_U$ [meV] | $E_{LE}$ [eV] | $E_{CT}$ [eV] | $\Delta E_{CT}$ [eV] |
|---|---|---|---|---|---|---|---|---|---|---|---|---|
| PM6:BO-EH-SCl | 1.484 | 1.216 | 1.136 | 0.926 | 0.268 | 0.08 | 0.21 | 0.558 | 18.8 | 1.484 | 1.390 | 0.094 |
| PM6:BO-EH-ACl | 1.484 | 1.216 | 1.134 | 0.932 | 0.268 | 0.082 | 0.202 | 0.552 | 19.5 | 1.484 | 1.420 | 0.064 |

chains. On the one hand, the symmetric NFAs with straight side chains showed significantly better charge generation properties than those of asymmetric NFAs. Despite the increase in $V_{oc}$, the magnitude of decreases in $J_{sc}$ for asymmetric NFA-based devices were much larger, resulting in lower PCEs for asymmetric NFAs with straight side chains compared with their corresponding symmetric NFAs. On the other hand, the improved charge dynamics properties resulted in improved $J_{sc}$ and FF, which coordinated assistance to the higher $V_{oc}$ to obtain higher efficiencies of asymmetric NFAs with all branched side chains compared to the symmetric ones. Finally, we found the best efficiency of D:A binary OPV is realized by the asymmetric NFAs, i.e. the PM6: BO-EH-ACl, which exhibits the best efficiency of 18.80%, with a high $V_{oc}$ of 0.928 V, $J_{sc}$ of 25.31 mA cm$^{-2}$ and FF of 80.11%. Notably, this result is among the best of D:A binary OPVs without any special treatment as shown in Supplementary Table 1, and validates the efficacy of asymmetric structure for high device performance.

## $V_{oc}$ loss analysis

To decipher the mechanism underlying the improved $V_{oc}$, we carried out the $V_{oc}$ loss analysis. According to previous work, the $V_{oc}$ loss is classified into three channels, which conform to the mechanism of charge recombination, namely the $\Delta V_1$, $\Delta V_2$, and $\Delta V_3$, as shown in the equation below and Fig. 2e and Table 2.

$$
\begin{aligned}
q\Delta V_{oc,loss} = E_g - qV_{oc} &= (E_g - qV_{oc}^{SQ}) + (qV_{oc}^{SQ} - qV_{oc}^{rad}) + (qV_{oc}^{rad} - qV_{oc}) \\
&= (E_g - qV_{oc}^{SQ}) + qV_{oc}^{rad,below\,gap} + qV_{oc}^{non-rad} \\
&= q\Delta V_1 + q\Delta V_2 + q\Delta V_3
\end{aligned}
$$

(1)

Where $\Delta V_1$ is the inevitable radiative recombination loss above the $E_g$, $\Delta V_2$ is the radiative recombination loss below the $E_g$, and $\Delta V_3$ is the non-radiative recombination loss. Since the band gap of the OPVs exhibits little difference between symmetric and asymmetric molecular structure, the $\Delta V_1$ shows the same value of 0.268 eV. The $\Delta V_2$ is relevant to the energetic disorder-induced absorbing tail structures, which can be quantified with the Urbach energy ($E_U$) via the highly sensitive EQE (sEQE) technique[22]. As shown in Supplementary Fig. 34, the calculated $E_U$ of the asymmetric BO-EH-ACl-based device is 19.5 meV, which is slightly larger than that of the symmetric BO-EH-SCl-based device as 18.8 meV, suggesting that the asymmetric structure may bring more energetic disturbance due to more diverse D:A interactions and thus slightly increases the radiative recombination loss below the $E_g$. Accordingly, the BO-EH-ACl-based OPV shows slightly higher $\Delta V_2$ (0.082 eV) than the asymmetric counterparts (BO-EH-SCl-0.080 eV). A similar observation has been reported by Tobin et al., that the voltage loss of the asymmetric acceptors is still lower than that of the symmetric counterparts[23], so the increased $\Delta V_2$ caused by the disorder may not affect the overall $V_{oc}$ loss reduction of asymmetric acceptors. The $\Delta V_3$ is highly relevant to non-radiative recombination loss and can be measured by the electroluminescence quantum efficiency (EQE$_{EL}$) of a device, according to the formula below[24]:

$$
q\Delta V_3 = q\Delta V_{oc}^{non-rad} = -kT\ln(EQE_{EL})
$$

(2)

Where $k$ is the Boltzmann constant, $T$ is the temperature in Kelvin. As shown in Fig. 2f, the based cell shows a higher EQE$_{EL}$ (3.912 × 10$^{-4}$) than

BO-EH-SCl-based cell (2.902 × 10$^{-4}$), leading to a lower $\Delta V_3$ of the former one as 0.202 eV than the latter one as 0.210 eV. Therefore, the asymmetric BO-EH-ACl-based OPV showed obviously smaller $\Delta V_3$ than that of the symmetric BO-EH-SCl-based counterpart. These results suggest that the asymmetric molecular geometry might be beneficial for inhibiting non-radiative decay, which contributes to the improvement in device performance. To confirm the above argument, we measured the EQE$_{EL}$ values of all other pairs of acceptors. As shown in Table 1, we find all asymmetric acceptors-based OPVs show higher EQE$_{EL}$ than their symmetric counterparts, and thus a reduced non-radiative $V_{oc}$ loss.

It is recognized that the charge recombination mainly occurs at the D:A interface and the energy of the charge-transfer state ($E_{CT}$) is directly relevant to the non-radiative charge recombination. To understand the mechanism of molecular symmetries on the non-radiative charge recombination, we extracted the $E_{CT}$ via the sEQE and normalized electroluminescence (EL) spectra (Fig. 2g, h). The $E_{CT}$ of asymmetric PM6:BO-EH-ACl-based blends is determined as 1.420 eV, which is higher than that of the symmetric PM6:BO-EH-SCl-based device (1.390 eV). The asymmetric NFAs exhibit a lower energy offset between the optical band gap ($E_g^{opt}$) and the charge-transfer state ($\Delta E_{CT} = E_g^{opt} - E_{CT}$, -0.064 eV) than their symmetric counterpart (0.094 eV), for their $E_g^{opt}$s are identical (1.484 eV) as shown in Supplementary Fig. 35 and Table 2. Lower $\Delta E_{CT}$ contributes to a stronger intensity borrowing effect or electronic coupling for higher luminescence[25], and thus a lower $V_{oc}$ loss for asymmetric molecular geometry. The $E_{CT}$ at D:A interface is related to the energetic structures and the coulombic force between the positive polaron at the donor and the negative polaron at the acceptor. Besides diverse D:A interactions exist in the symmetric-asymmetric NFA pairs-based blends, it can be inferred that the other difference in $E_{CT}$ originated from their coulombic interactions, which is a function of D:A interfacial molecular interactions.

Besides charge recombination, the charge generation process is also demonstrated to be greatly related to the $E_{CT}$, because the $\Delta E_{CT}$ is the driving force for exciton dissociation. Therefore, the higher $E_{CT}$ also contributes to the slightly lower charge generation of the asymmetric NFA-based OPV, and thus, a lower $J_{sc}$ for asymmetric NFA.

## Charge transfer, transport, and recombination

To further understand the mechanism underlying the optoelectronic property variations between the symmetric and asymmetric acceptors, we studied their carrier dynamics, i.e. the charge transfer, transport, and recombination properties. First, the photoluminescence (PL) spectroscopy was measured as shown in Supplementary Fig. 36 and related quenching efficiency were summarized in Supplementary Table 4. The PL emission of all these NFAs can be quenched by over 92% efficiency after blending with the donor, i.e. PM6. While compared to the asymmetric binary blends, all the symmetric binary blends showed slightly higher PL quenching efficiency, indicating more effective charge transfer from NFAs to PM6, which is consistent with the higher charge generation driving force and better photocurrent.

Further, the charge-transfer dynamics were further investigated by femtosecond transient absorption spectroscopy (TAS) as shown in Supplementary Fig. 37 and Fig. 3. An 800 nm excitation light was used here to selectively excite the acceptors in the blend films. The primary excitation of both BO-EH-SCl and BO-EH-ACl is observed with a ground-state bleaching (GSB) signal at around 810 nm in the blend

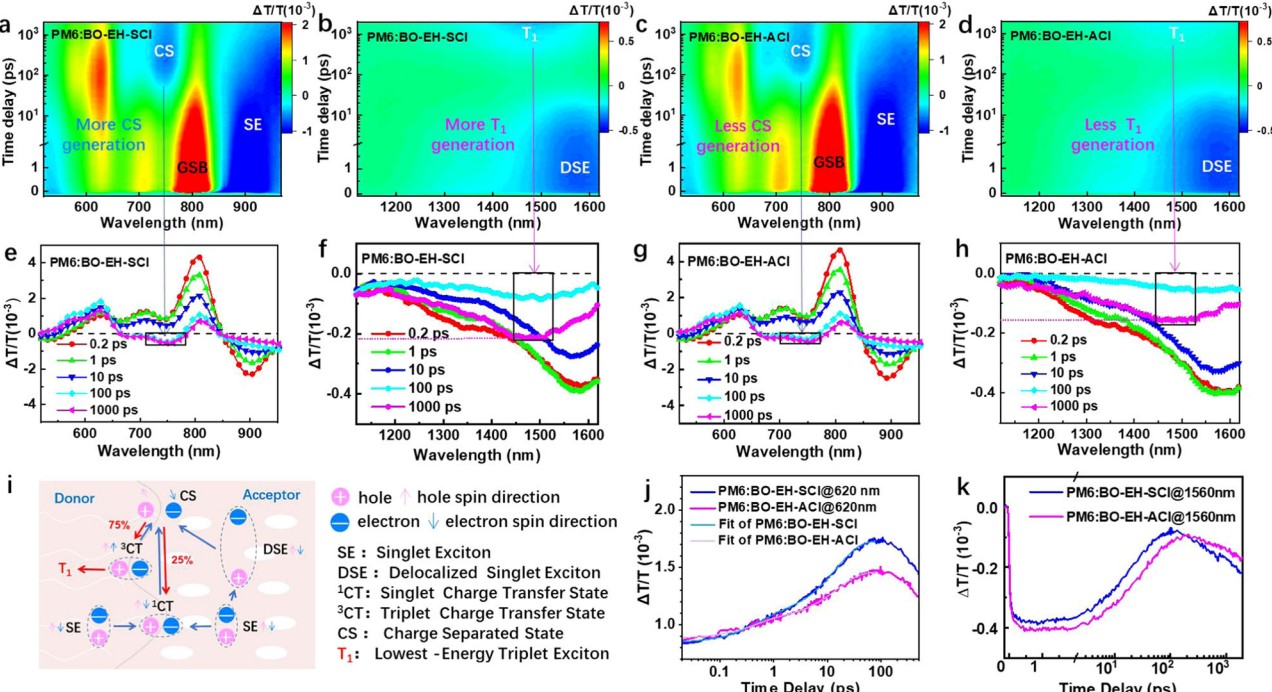

**Fig. 3 | Charge-transfer dynamics.** Representative 2D TAS images of **a**, **b** PM6:BO-EH-SCl and **c**, **d** PM6:BO-EH-ACl with pump at 800 nm. **e**–**h** The intensity profiles of the TAS spectra with varied time delays. **i** Schematic of the excited-state kinetics (Red lines or arrows represent the recombination routine). Hole transfer dynamics in two blend films **j** at 620 nm and **k** at 1560 nm.

films, which agrees with the film absorption spectra. The excited-state absorption (ESA) signals at around 900 nm for both blends represent the kinetic process of the localized exciton (LE) or singlet excited (SE) of NFAs. With the GSB and ESA signal peaks of acceptors (BO-EH-SCl and BO-EH-ACl) decreasing in Fig. 3e–h, a bleaching signal near the absorption peak of PM6 emerged at 620 nm, indicating hole transfer from the HOMO level of acceptors to HOMO level of PM6 as CT exciton. After around 100 ps, the signal peak of PM6 starts to decay and subsequently an absorption signal peak at 750 nm shows a rising behavior, indicating the appearance of PM6 hole polaron or charge separation (CS) state. Here, GSB kinetics of PM6 were utilized on behalf of hole transfer dynamics, which mainly adopted interfacial CT-state dissociation as shown in Fig. 3i. The lifetime of charge separation at D:A interface and charge diffusion process are derived to be $\tau_1$ and $\tau_2$ via bi-exponential fitting (Fig. 3j). The PM6:BO-EH-SCl and PM6:BO-EH-ACl blends exhibited the lifetimes ($\tau_1$, $\tau_2$) of (0.48 ps, 14.6 ps) and (0.43 ps, 14.8 ps), respectively, indicating negligible differences on charge transfer between the two blends via CT-state dissociation[11]. These results confirm the high charge generation efficiency for both symmetric and asymmetric NFAs, which is consistent with the high EQE spectra for both cases.

The coulombic attraction between positive and negative charges could be significantly reduced via exciton delocalization[26,27]. Due to the compact molecular stacking, the exciton separation will occur through the D:A interface, while the excitons can also spontaneously delocalize in pure NFA phase[27]. That is, the SE of acceptors at around 900 nm in Fig. 3a, c (or at around 910 nm as shown in Supplementary Fig. 37a, c in pure films) could be transferred into the delocalized singlet exciton (DSE, which exhibit low exciton binding energy with electron and hole on separate molecules) at around 1580 nm in Fig. 3b, d (or at around 1570 nm as shown in Supplementary Fig. 37b, d in pure films). The generation of DSE tends to facilitate the charge generation, which is evidenced by the formation of CS state at 750 nm (as shown in Fig. 3a, c) in the NFA phase of blends as earlier reported[28,29]. TAS traces of two blend films at 1560 nm with varied time delays were probed to

investigate the hole transfer process (Fig. 3k). The DSE lifetimes in the PM6:BO-EH-SCl and PM6:BO-EH-ACl blend film are 19.4 and 32.4 ps, respectively. The longer DSE lifetime of PM6:BO-EH-ACl blends is consistent with the reduced non-radiative charge recombination. For the BO-EH-SCl and BO-EH-ACl pure film, the DSE lifetimes are 185 and 167 ps. We assumed that most holes in the acceptor phase were obtained from DSE as reported work and ideally the DSE dissociate into CS immediately[29], so the hole transfer efficiency (HTE) via the DSE channel can be determined to be 89.5% and 80.6% for PM6:BO-EH-SCl and PM6:BO-EH-ACl systems, which is in accordance with higher photocurrent generation in symmetric NFAs.

Interestingly, a phenomenon was observed in the TAS spectra of blend films within the time scale of 100–1000 ps. In the first 100 ps, the hole transferred from the SE (at 900 nm) and DSE (at 1580 nm) of acceptors to PM6 (at 620 nm). After 100 ps, the stronger absorption intensity of free charge carrier for PM6:BO-EH-SCl blends occurred at 750 nm (Fig. 3b, d, f, h), indicating more CS state generated than PM6:BO-EH-ACl blends. However, the intensity of the CS states nearly retained from 100 to 1000 ps in both blends, but an absorption signal at around 1480 nm increased subsequently. In terms of time scale, it is in line with the spectral feature of the lowest-energy triplet exciton ($T_1$) in NFAs, which can irreversibly induced the non-geminate charge recombination loss[28,30]. As shown in Fig. 3i, SE→$^1$CT→CS is the main efficient photogenerated current channel via D:A interface. The non-geminate charge recombination from the CS state will eventuate the $^3$CT and $^1$CT state with a quantity ratio of 3:1. The decay rate of the $^3$CT state is much higher than that of $^1$CT, so the $^3$CT state mediates the non-radiative losses[30,31]. The kinetics from the $^3$CT to $T_1$ state is an important non-emissive decay pathway, which will increase the non-radiative voltage loss. The rate constant of non-radiative decay between two molecular electronic states can be described by Marcus' equation derived via the Fermi Golden Rule, which suggests that a larger energy gap between $^3$CT and T1 suppresses $^3$CT→T1 rate constant[31]. The $^3$CT state usually has a higher energy level than the T1 state. Under the assumption that the approximate T1 energy level

for two similar NFAs, it is reasonable to understand that the higher CT energy of asymmetric NFAs allowed for less production of triplet excitons with lower occurrence probability for the $^3$CT→T1 process. Compared to PM6:BO-EH-ACl blends, the $T_1$ signal for PM6:BO-EH-SCl blends shows much stronger absorption intensity at 100–1000 ps time scale under the same excitation fluence, implying more $T_1$ generation. Therefore, these results indicate that the asymmetric molecular geometry is capable of suppressing $T_1$ formation, which results in less non-radiative voltage loss and it is in accordance with the lower $\Delta V_3$[32,33].

With regard to charge transport, the hole and electron mobilities ($\mu_h$ and $\mu_e$) of the acceptors were measured with the space-charge-limited-current (SCLC) mechanism as shown in Supplementary Fig. 38 and Supplementary Table 4. We observe that the asymmetric acceptors (2BO-ACl and BO-EH-AX (X = Cl, Br)) deliver slightly improved hole and electron mobilities than their symmetric counterparts (2BO-SCl and BO-EH-SX (X = Cl, Br)). For example, BO-EH-SCl-based devices showed $\mu_h$ and $\mu_e$ values of $2.90 \times 10^{-3}$ and $5.08 \times 10^{-3}$ cm$^2$ V$^{-1}$ s$^{-1}$ with $\mu_e/\mu_h$ as 1.75. While the BO-EH-ACl-based devices showed $\mu_h$ and $\mu_e$ values of $3.12 \times 10^{-3}$ and $5.25 \times 10^{-3}$ cm$^2$ V$^{-1}$ s$^{-1}$ with lower $\mu_e/\mu_h$ as 1.68. Asymmetric acceptors BO-EH-ACl-based devices showed higher and more balanced charge mobility than that of symmetric acceptors BO-EH-SCl, which is favorable to enhancing effective charge collection and thus increase FF. While we find the side chains play a more significant role in determining the carrier mobility. In general, the NFAs with all branched side chains, i.e., the 2BO-SCl, 2BO-ACl, BO-EH-SX, and BO-EH-AX (X = Cl, Br) exhibit better charge transport properties than those with branched two straight side chains, i.e. the C11-BO-SX and C11-BO-AX (X = F, Cl, Br). Therefore, the balance among charge generation, charge recombination, and charge transport determines the best performance of OPV achieved with asymmetric molecules of branched side chains.

**Electrostatic potential (ESP), Molecular-dynamics (MD) simulations, and TDDFT calculations**

As shown, the detailed molecular packing and interaction structures at the D:A interfaces are the key to understanding the role of molecular asymmetry on the carrier dynamics and device performance. Since GIWAXS technology commonly used to investigate OPV blend film morphologies presents morphological characteristics typically at the scale of tens of nanometer, direct experimental characterizations of detailed molecular packing and interaction structures at the D:A interfaces are lacked. Here, we carried out all-atom molecular-dynamics (AA-MD) simulations (for more details, see "Supplementary Information Molecular-Dynamics Simulations Method" section) to unveil the working mechanism of asymmetric geometry at molecular level[12,34], and Source data are provided as a Source Data file. We selected the BO-EH-SBr and BO-EH-ABr symmetric-asymmetric pair as our objective. The end groups on the two sides of the NFA are labeled as A$_1$ and A$_2$. For symmetric NFA, A$_1$ is the same as A$_2$ as 2-(5 or 6-bromo-3-oxo-2,3-dihydro-1H-inden-1-ylidene)malononitrile (IC-Br). In one case, the A$_1$ is 2-(5,6-dibromo-3-oxo-2,3-dihydro-1H-inden-1-ylidene)malononitrile (IC-2Br) and A$_2$ is 2-(3-oxo-2,3-dihydro-1H-inden-1-ylidene)malononitrile (IC) as shown in Fig. 1a. Before conducting the AA-MD, we calculated the Electrostatic potential (ESP) of five relevant fragments in the D:A interfaces as reference. As shown in Fig. 4a, the D$_0$ fragments of donor PM6 display a negative ESP value on most of the molecular surface, indicating its electron-rich nature. In contrast, the ESP values are positive on most of the IC-XBr (X = 0, 1, 2) and A$_0$ of donor PM6 surfaces, suggesting an electronic affinity. The ESP values of every atoms in the five fragments were further given in Supplementary Table 5. The atoms in the D$_0$ moiety showed obviously average negative ESP values compared to the A$_0$ moiety, indicating a higher ESP gap of the former with the IC-XBr (X = 0, 1, 2) fragments in the D/A interfaces. The increased ESP difference between donor and

acceptor will enhance the intermolecular interaction, suggesting stronger intermolecular interaction for the D$_0$ and IC-XBr (X = 0, 1, 2) fragments compared to the counterpart of A$_0$ and IC-XBr (X = 0, 1, 2) fragments. With more bromine atoms, most atoms on IC-XBr (X = 0, 1, 2) showed higher ESP values because of the electron-withdrawing properties of the bromine atom. However, the bromine atoms at the bottom of the IC-XBr (X = 0, 1, 2) labeled as 18 and 19 showed significantly decreased ESP value (Fig. 4c), probably attributed to the larger electronegativity ($\chi$) value of bromine (2.8) than hydrogen (2.1) and carbon (2.5) atoms[35]. It's worth noting that the 18th and 19th atoms on IC displayed the highest ESP values among all the atoms on the IC-XBr (X = 0, 1, 2), leading to the highest dipole moment as 4.66 D among the three end groups of NFAs (IC-Br: 4.03 D; IC-2Br: 3.14 D). The larger dipole moment of the end group will lead to a stronger intermolecular interaction with the PM6 donor, especially the D$_0$ moiety. The interaction energy between the end group of NFA and the PM6 donor (classified into the D$_0$ and A$_0$ moieties) was calculated, where four representative molecular interaction structures are selected, i.e. the D$_0$-A$_1$, D$_0$-A$_2$, A$_0$-A$_1$, and A$_0$-A$_2$. As shown in Fig. 4b, the asymmetric NFA naturally exhibits more diverse D:A interfacial molecular structures compared to the symmetric structure. Moreover, we find the symmetric NFA exhibits similar interaction energy for the four molecular interaction structures. While, for the asymmetric NFA, the D$_0$-A$_2$ interaction structure exhibits much higher interaction energy, which determines it the favorable interfacial structure (Fig. 4d). Adjusting the fragments for PM6 with the same atomic numbers, the optimal molecular structure was maintained as Supplementary Fig. 39. As a result, we find from the MD simulation that the D$_0$-A$_2$ is the dominant molecular interaction structure at the microscopic structure of D:A interface. The greater interaction further leads to higher contact probability between A$_2$ and D$_0$ for asymmetric NFAs, displayed in Fig. 4e. Further, the $E_{CT}$s of different molecular interaction structures are studied. A big number of D:A complexes, which contain the pairs formed by end group A$_1$ of BO-EH-SBr with donor PM6, end group A$_1$ of BO-EH-ABr with donor PM6, and end group A$_2$ of BO-EH-ABr with donor PM6, were extracted from the last 10 ns of MD trajectories, and were performed with DFT calculations to characterize the nature of CT states. We calculated HOMO and LUMO of three typical complexes (D$_0$-A$_1$ or A$_2$ for PM6:BO-EH-SBr, A$_0$-A$_1$ and A$_0$-A$_2$ for PM6:BO-EH-ABr displayed in Supplementary Fig. 40) extracted from the MD simulation box, and our results demonstrated that HOMO and LUMO wavefunctions distribute on the donor PM6 and NFA acceptor, respectively, implying that the first excited singlet states in these complexes have CT nature. For A$_1$ of BO-EH-SBr and PM6 complex, the CT energy averaged on all D:A complexes extracted from MD simulations is 1.196 eV (Fig. 4f). Interestingly, the average CT-state energy for the extracted A$_2$ of BO-EH-ABr and PM6 complexes is higher, about 1.214 eV (Fig. 4g), although the average CT-state energy for the extracted A$_1$ of BO-EH-ABr and PM6 complexes is about 1.159 eV (Fig. 4h). Importantly, our MD simulation results indicated that A$_2$ of BO-EH-ABr and PM6 conformation with lower energy has a larger probability, implying that the CT states for A$_2$ of BO-EH-ABr and PM6 complexes play a dominant role in exciton dissociation and recombination. Due to higher average CT-state energy for A$_2$ of BO-EH-ABr and PM6 conformation, the CT-state non-radiative recombination rate would be reduced according to energy-gap law, and, in addition, the CT-LE energy gap at such interfaces is also reduced, which is beneficial to a larger EQE$_{EL}$ (i.e., a smaller non-radiative voltage loss) (Tables 1 and 2). Meanwhile, it is also conducive for asymmetric NFAs to achieve a lower energetic offset between the Energy band gap ($E_g$) and the charge-transfer state ($\Delta E_{CT} = E_g - E_{CT}$) than the symmetric counterpart. Lower $\Delta E_{CT}$ was adverse to efficient charge separation and thus less efficient charge generation, which is consistent with the experimental results.

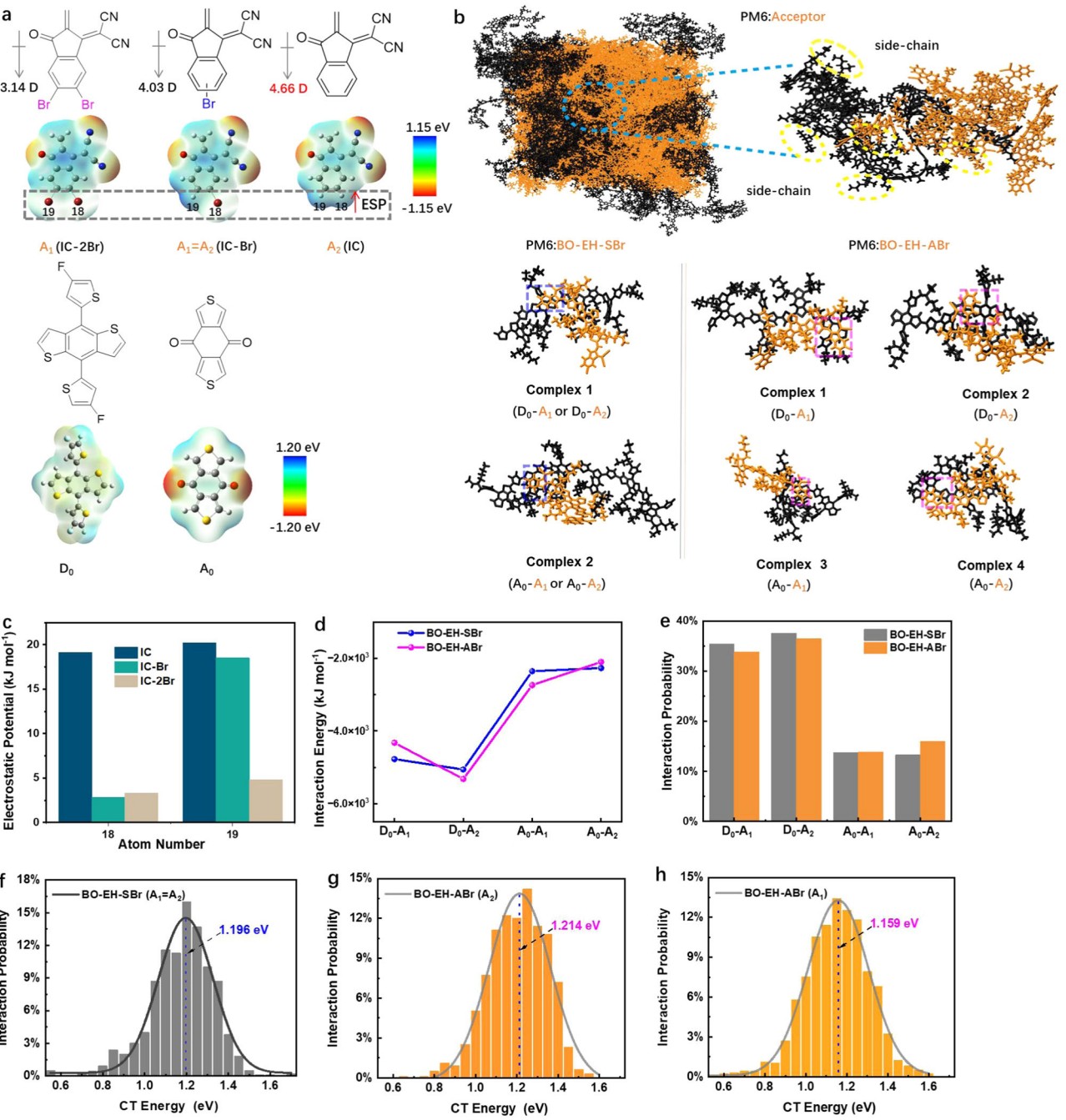

**Fig. 4 | The electrostatic potential (ESP), Molecular-dynamics simulations, and TDDFT calculations for PM6:BO-EH-SBr and PM6:BO-EH-ABr complexes. a** The ESP distribution images of five fragment structures. **b** Intermolecular stacking configurations of the complexes with side chains. **c** ESP values of two atoms on three fragments. **d** Interaction energy, and **e** contact probability. **f**–**h** The distribution of CT energy of donor PM6 with end groups (A₁, A₂) of different acceptors.

## Discussion

The microscopic structures and energetics at the D:A interface are the key to determining the optoelectronic properties of bulk heterojunction OPV. The current researches in the field of OPV tend to intensively study how to manage the energetics at the D:A interface via fine-tuning the energy levels of either donor or acceptors, which also dramatically affect device performance. A general rule has been outlined: the smaller energetic offset at the D:A interface tend to suppress the nonradiative recombination, while the larger energetic offset will facilitate the charge generation. This determines the compromise between the $J_{sc}$ and $V_{oc}$, and thus the overall efficiency. In the fullerene acceptor generation, researchers find that the D:A interfacial stacking modes, e.g. the "edge-on" and "face-on" of pentacene on fullerene, is an

important factor governing the device performance of OPV[36]. With the revolution of materials, the molecular interaction structures or the molecular conformations at the D:A interfaces become even more complex, which should also play a critical role. Therefore, the control over the D:A interfacial molecular structure or conformation might provide the possibility to the increased $V_{oc}$ without sacrificing $J_{sc}$. However, there is few work focusing on this topic, and the corresponding strategies to manipulate the D:A interfacial conformations are severely lacked. For the NFA with end group asymmetry, it will naturally generate more diverse D:A interfacial molecular structures. Our previous work observed that manipulating the CT energy via asymmetric structure can affect the photon-to-electron conversion. However, the accompanying variation in energetic structure with

changing the molecule from symmetric to asymmetric makes it difficult to disentangle the effect of the molecular geometry and the energetic structures. Via the appropriate molecular design, we demonstrated that the asymmetric NFAs deliver more diverse D:A interfacial structures, and this results in higher CT energy and lower triplet state density, which is the key to the suppressed non-radiative voltage loss.

More detailed impact of $E_{CT}$ and triplet exciton formation on $V_{oc}$ loss can be comprehended via the approximate steady-state equilibrium between the reformation of CT states (via $T_1$ dissociation into CT states) and $T_1$ decays, as described by Vandeval et al.[37]. Herein, $V_{oc}$ can be quantified as below:

$$V_{oc} \approx \frac{\Delta E_{CT}}{q} - \frac{kT}{q} \ln\left(\frac{k_R(CT) + k_{CT-T}}{G} N_{CTC}\right) \quad (3)$$

Where $k_R(CT)$ is the back transfer[30] rate from $T_1$ to $^3CT$, and $k_{CT-T}$ is the energy transfer rate from $^3CT$ to $T_1$. $G$ and $N_{CTC}$ represent charge carrier photogeneration rate and the total density of CT complexes in blends respectively. Simplistically, the $N_{CTC}$ and $G$ for PM6: BO-EH-SCl and PM6: BO-EH-ACl blends is assumed to be similar. In general, the T1 is observed in PM6:Y6-derivative because $k_R(CT) \ll k_{CT-T}$[30,33]. Therefore, we conclude that the $V_{oc}$ difference ($\Delta V_{oc}$) between PM6: BO-EH-SCl and PM6: BO-EH-ACl blends can be attributed to different $E_{CT}$ and triplet exciton formation rate as follows:

$$\Delta V_{oc} \approx \frac{\Delta E_{CT}}{q} - \frac{kT}{q} \ln(k_{CT-T}) \quad (4)$$

On the one hand, the $E_{CT}$ of PM6:BO-EH-ACl blends (1.42 eV) was obviously larger than PM6:BO-EH-SCl blends (1.39 eV) as shown in Fig. 2g, h. On the other hand, more feeble $T_1$ absorption was detected by TAS in Fig. 3b, d, indicating smaller $k_{CT-T}$ value of PM6:BO-EH-ACl blends than PM6:BO-EH-SCl blends. Therefore, enhancing the CT energy via D:A interface and inhibiting the $T_1$ formation were the two main reasons for PM6:BO-EH-ACl blends to achieve larger $V_{oc}$.

In summary, we have demonstrated that the molecular asymmetric geometry plays a critical role in determining the device performance, especially the non-radiative energy loss, via the appropriate design of twelve acceptor molecules through DFT calculations and experiments. As a result, the asymmetric acceptors-based devices universally exhibit higher open-circuit voltage ($V_{oc}$) than that of the symmetric acceptors. Due to the suppressed non-radiative recombination loss via high CT energy and $T_1$ inhibition, the asymmetric acceptors BO-EH-ACl-based OPV finally reaches the best efficiency of 18.80%. Our work has revealed the impact of D:A molecular interaction structures on the CT energies for the asymmetric NFAs via AA-MD simulations, and their association with the non-radiative charge recombination and triplet dynamics, which provides a novel perspective to understand the working mechanism of OPV and should inspire a future avenue of tailoring the D:A interfacial structure for high-performance OPVs.

## Methods

### Materials
The donor PM6 was purchased from Solarmer Materials Inc. The detailed synthesis routines of all the acceptors are described in Supplementary Methods.

### Device fabrication
Organic photovoltaics were fabricated on glass substrates commercially pre-coated with a layer of indium tin oxide (ITO) with the conventional structure of ITO/PEDOT:PSS/active layer/PDINN/Ag. Before fabrication, the substrates were cleaned using detergent, deionized water, acetone, and isopropanol consecutively for 10 min in each step. And then the ITO substrates were treated in the ultraviolet ozone generator (UC100-SE, LEBO Science) for 20 min before being spin-coated at 4500 rpm with a layer of 15 nm thickness PEDOT:PSS (Clevios 4083). After baking the PEDOT:PSS layer in air at 150 °C for 10 min, the substrates were transferred to the $N_2$ glovebox. The D:A ratio is 1:1.2 wt% for all blends, and the total concentration is 16 mg/mL. Heat and stir at 60 °C for 1 h before spin coating. Then an annealing at 90 °C for 5 min was performed. A thin layer of PDINN was spin-coated from 1.5 mg/mL methanol solution on the top of the active layer. Finally, the Ag (100 nm) electrode was deposited by thermal evaporation to complete the device with an active area of 6 mm$^2$, and the testing aperture area was 4.572 mm$^2$.

### J–V and EQE measurements
The J–V measurement was performed via the solar simulator (SS-X50, Enlitech) and AM 1.5G spectra, calibrating the intensity of the certified standard silicon solar cell (KG2) at 100 mW·cm$^{-2}$. The external quantum efficiency (EQE) data were obtained using the solar-cell spectral-response measurement system (RE-R, Enlitech).

### AA-MD simulation
The MD simulations were performed using the GROMACS package[38] with the optimized potentials for liquid simulations-all-atom (OPLS-AA) force field[39,40]. OPLS-AA is one of the most popular all-atom force fields widely used in studies of organic molecules. For all the MD simulations, periodic boundary conditions were applied, and to simulate the experimental annealing process, MD simulations were performed from 353 (30 ns) to 300 K (30 ns), where 353 K is the annealing temperature, with a cooling rate from 353 to 300 K of 10 K/ns. The LINCS algorithm was applied to constrain the covalent bonds with H-atoms[41]. The time step of the simulations was 1.0 fs. The cut-off of the non-bonded interactions was set to 12 Å. The particle mesh Ewald (PME) method was used to calculate the long-range electrostatic interactions[42]. The graphics and visualization analyses were processed by the Visual Molecular Dynamics (VMD) program[43]. All the Density Functional Theory (DFT) calculations were performed using Gaussian 16[44].

### Reporting summary
Further information on research design is available in the Nature Portfolio Reporting Summary linked to this article.

## Data availability
Source data are provided with this paper. The authors declare that all relevant data are included in the paper and its Supplementary Information. Source data are provided with this paper.

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

## Acknowledgements

This work is supported by the National Natural Science Foundation of China (NSFC, Nos. 52173185, 52127806, 21875182, 52303247, and 52173023), and the National Key Research and Development Program of China (Nos. 2022YFB4200600, and 2022YFE0132400), Key Scientific and Technological Innovation Team Project of Shaanxi Province (2020TD-002), and 111 project 2.0 (BP0618008). L.Z. thanks the research start-up fund from Zhejiang University. Thanks for the support from the X-ray data was acquired at beamlines 7.3.3 at the Advanced Light Source, which is supported by the Director, Office of Science, Office of Basic Energy Sciences, of the U.S. Department of Energy at Lawrence Berkeley National Laboratory under Contract No. DE-AC02-05CH11231. The authors thank Dr. Eric Schaible and Dr. Chenhui Zhu at beamline 7.3.3 for assistance with data acquisition. X.-K.C. also gratefully acknowledges the financial support from Suzhou Key Laboratory of Functional Nano & Soft Materials, Collaborative Innovation Center of Suzhou Nano Science & Technology, and the 111 Project.

## Author contributions

L.Z. (Lijian Zuo) and J.H. conceived the idea. J.H. synthesized the non-fullerene acceptors and performed the related characterizations. Y.P., T.C., and M.W. fabricated the OPV devices and performed the related measurements. Y.Z. and Z.M. carried out the energy loss test. C.Z., H.Z., J.C. and Y.O. carried out TRPL and TAS characterizations. Z.B. and W.M. did the GIWAXS measurements. X.C. and L.M. carried out TDDFT calculations and AA-MD simulation. L.Z. supervised the project. The manuscript was mainly written by J.H., H.C. and L.Z., and all authors commented on the manuscript.

## Competing interests

The authors declare no competing interests.
