## [Peer Review File · Nature Communications]

On the role of asymmetric molecular geometry in high-performance organic solar cellsREVIEWER COMMENTS

Reviewer #1 (Remarks to the Author):

In this manuscript, Zuo and coworkers synthesized six pairs of symmetric and asymmetric norfullerene acceptors (NFAs) by varying the halogen atoms (F, Cl and Br) in the end-groups. The asymmetric NFAs exhibited identical optoelectronic properties to their respective symmetric counterparts. The authors successfully fabricated the conventional type of organic solar cells (OSCs) by blending these NFAs with the PM6 polymer donor. Despite similar optical and electrochemical properties, the PM6:NFA-based OSCs demonstrated distinct photovoltaic properties. Their findings revealed that OSCs based on asymmetric NFAs consistently achieved higher open-circuit voltage than their symmetric counterparts, attributed to reduced non-radiative recombination. Molecular-dynamic simulations unveiled that asymmetric NFAs displayed more diverse molecular interaction patterns at the donor: acceptor interface and a higher donor: acceptor interfacial charge-transfer state energy. Moreover, the studies clarified that the asymmetric structure effectively inhibited formation of triplet state. Consequently, the authors achieved a superior power conversion efficiency of 18.80%, primarily due to lower non-radiative energy loss. Although the comparative study between symmetric and asymmetric NFAs in this work has been done well, novelty seems lack in terms of the chemical structures and the idea of an asymmetric strategy. Also, the manuscript contains many flaws. It is recommended that the authors need to address several issues listed below for the improvement of the manuscript quality.

Comments:

1. Why are the HOMO and LUMO energy levels of 2BO-EH-based NFAs (2BO-EH-SCI and 2BO-EH-ACI) significantly downshifted compared to 2BO-based NFAs (2BO-SCI and 2BO-ACI), despite having a similar conjugated backbone? Similarly, why did 2BO-based NFAs-based devices exhibit lower efficiencies compared to 2BO-EH-based NFAs-based devices?
2. Figure S35 shows that there were negligible differences in photoluminescence (PL) quenching in both asymmetric and symmetric binary blends. But, the authors claimed that the asymmetric binary blends exhibited a higher level of PL quenching compared to the symmetric binary blends. It is not understandable. Please provide appropriate comments on this or revise it.
3. Despite an increase in Voc, straight side chain appended asymmetric NFAs-based OSCs exhibited lower efficiencies compared to their symmetric counterparts. What is the limiting factor?
4. Why were C11-BO-SBr and C11-BO-ABr, along with their blends, chosen for GIWAXS analysis over higher-performing blends (PM6:BO-EH-SCI and PM6:BO-EH-ACI)? The GIWAXS data related to PM6:BO-EH-SCI and PM6:BO-EH-ACI should be included in the manuscript.
5. The reference 14 in the reference section is incorrect, please check and correct it. It should be "Chem. Sci., 2021, 12, 14083–14097"
6. The HOMO and LUMO labeling in Figure S1 is incorrect, please correct them.
7. The LUMO value of ACI is missing in Figure S2, include it.

8. In line 206 on page 10 in the manuscript, it is written as 'symmetric and symmetric pairs can be derived...' and should be corrected to 'symmetric and asymmetric pairs can be derived...
9. In the Figure captions, please write Figure 30 as Figure S30 and Figure 31 as Figure S31..
10. The authors did not mention Figure S33, S37 and S39 in the main text of the manuscript.
11. The contact angles of the NFAs do not match between Figure S30 and Table S2, except for PM6.
12. The synthesis of C11-BO-SCl, 2BO-SCl and BO-EH-SCl in the supporting information is written as “2-(5 or 6-dichloro-3-oxo-2,3-dihydro-1H-inden-1-ylidene)malononitrile (61 mg, 0.27 mmol)” and should be corrected by removing ‘di’ to “2-(5 or 6-chloro-3-oxo-2,3-dihydro-1H-inden-1-ylidene)malononitrile (61 mg, 0.27 mmol).

Reviewer #2 (Remarks to the Author):

In this work present by Jinfeng Huang, Tianyi Chen, Le Mei et al. the role of asymmetric molecular geometry on the performance of organic photovoltaics (OPV) was reported. By designing and synthesizing six pairs of symmetric and asymmetric non-fullerene acceptors with identical physical and optoelectronic properties, the authors found that asymmetric acceptors exhibit higher open-circuit voltages and lower non-radiative charge recombination compared to symmetric counterparts. Molecular dynamics simulations revealed asymmetric acceptors have more diverse donor-acceptor orientations and higher interfacial charge transfer state energy, reducing non-radiative decay. The asymmetric acceptor BO-EH-ACl based device achieved a high efficiency of 18.80%. Authors highlight the uniqueness of molecular asymmetry for high-performance OPV by manipulating donor-acceptor interfaces to balance carrier dynamics.

Overall, we feel that this work is not suitable for publication in Nature Communications for the following reasons.

1. Reading through the manuscript, it is so hard to understand what scientifically new findings or new conclusions are claimed in the work. Basically, it is already well known that asymmetric NFAs with different EGs on each side of the molecule exhibit lower charge recombination probability resulting from the higher triplet state energy. So many papers have made such claims using the methods of electroluminescence-EQE and transient absorption. The same ideas have been reported many times since the first ITIC-type NFA.
2. The high performance of 18.80% PCE proposed in this study is not something new either for OSC devices. 18+% PCE has been reported since the beginning of 2022. Fabricating OSC devices with higher performance is good, but essentially it is an engineering problem, or to be specific, nowadays it is a full factor engineering problem. The scientific part of such research study should focus on the fundamentals behind such high performance. The discoveries proposed in the study should allow readership in the community to develop similar or even higher-performance devices. Repeating the similar performance

of devices by testing different combinations of similar D/A materials (or end group combinations) does not advance this field.

3. In the work, the authors use DFT-based molecular dynamic simulation to study the interactions between the donors and NFAs. The authors proposed that asymmetric NFA exhibits more diverse interactions with the donor phase with a D0-A2 orientation as dominant. However, there are no experimental data to support this statement. The GIWAXS data listing in the supporting information is the same for the symmetric and asymmetric blends. There is no significant diffraction signal that can prove that the claimed orientation only exists in asymmetric NFA blends. And there is no experimental evidence to suggest that there are more diverse D-A interactions. In taking a close look at the computational results solely, it can be seen that all the alkyl chains from both the donor and acceptor were either removed or simplified to the methyl group in the simulation. But in the main text of the manuscript, the authors claimed that they conducted an 'All-Atoms Molecular Dynamics Simulation (AA-MD)'. So, what the authors mean here for 'all atoms' is clearly incorrect. More importantly, it is already so well known that the alkyl chains of the materials play an important role in determining the final morphology of -A BHJ blends. Many high-performance materials that were developed based on Y6 originate solely from the manipulation of the alkyl chains of Y6, for example, the BTP-EC9 and L8-BO. Thus, it is unacceptable to study the donor-acceptor interaction in simulation without considering the influence of the alkyl chains and thus trying to derive key conclusions of the manuscript based on such unsupported simulation results.

Reviewer #3 (Remarks to the Author):

In this manuscript, the authors studied the independent role of molecular asymmetry on the device performance of OPVs. They have designed and synthesized six pairs of symmetric-asymmetric non-fullerene acceptors (NFAs), which exhibited identical physical and optoelectronic properties for each symmetric and asymmetric NFA pair. Compared to their symmetric counterparts, these asymmetric NFAs also demonstrated increased open-circuit voltage and lower non-radiative charge recombination loss. With carrier dynamic measurements and MD simulations, the authors unveiled that the asymmetric NFA based OPV exhibit more diverse D:A interfacial structures with averagely higher charge transfer states, and less T1 formation, which directly contributes to the reduced non-radiative charge recombination. As a result, best device performance is reached with asymmetric NFAs, with power conversion efficiency of 18.8%, and is one of the highest efficiency among binary OPVs. This work illustrate the working mechanism of molecular asymmetry on the device performance of OPVs, provides a new perspective of manipulating the D:A interfacial contact structures, and should have important implications on molecular structure design for high-performance OPVs. Therefore, the reviewer recommends this work to be published in Nat. Commun. after addressing the following issues.

1. As mentioned in this paper, both higher CT energy and T1 suppression are conducive to inhibit the non-radiative voltage loss. The link between the asymmetric molecular structure and the T1 formation is weak, and this part should be discussed in more details.

2. The D:A interfacial contact structures is very important for optoelectronic properties of OPVs. Besides of the charge recombination, I believe this will also dramatically affect the charge generation process. Moreover, the asymmetric structures tend to form a interfacial dipole moments, and I suggest the authors discuss their effects on the charge generation.

3. Although there is a reliable variation trend on the short circuit current density (J_{sc}) and V_{oc} values between symmetric and symmetric pairs as Figure 2b shown, the PCE variations for acceptors with straight side chains, like C11-BO-SX and C11-BO-AX (X=F, Cl, Br), were totally different from the NFAs with all branched side chains, like 2BO-SCl and 2BO-ACl, BO-EH-SX and BO-EH-AX (X= Cl, Br). This part should be discussed in more details.

4. The authors studied the light intensity dependent voltage and photo-current density, and the corresponding results are presented in Figure S38 in the charge transfer, transport and recombination part of the supporting information. However, there are no relevant results and discussions provided in the main text.

5. Figures in this work should be polished. For example, the Figure 1b contains two figures with similar information, and the illustrations inside the figures is not concise.

6. Since end group tailoring is the key molecular design strategy in this work, more relevant references should be cited.

Response letter

Reviewers' comments:

Reviewer #1 (Remarks to the Author):

In this manuscript, Zuo and coworkers synthesized six pairs of symmetric and asymmetric norfullerene acceptors (NFAs) by varying the halogen atoms (F, Cl and Br) in the end-groups. The asymmetric NFAs exhibited identical optoelectronic properties to their respective symmetric counterparts. The authors successfully fabricated the conventional type of organic solar cells (OSCs) by blending these NFAs with the PM6 polymer donor. Despite similar optical and electrochemical properties, the PM6:NFAs-based OSCs demonstrated distinct photovoltaic properties. Their findings revealed that OSCs-based on asymmetric NFAs consistently achieved higher open-circuit voltage than their symmetric counterparts, attributed to reduced non-radiative recombination. Molecular-dynamic simulations unveiled that asymmetric NFAs displayed more diverse molecular interaction patterns at the donor: acceptor interface and a higher donor: acceptor interfacial charge-transfer state energy. Moreover, the studies clarified that the asymmetric structure effectively inhibited formation of triplet state. Consequently, the authors achieved a superior power conversion efficiency of 18.80%, primarily due to lower non-radiative energy loss. Although the comparative study between symmetric and asymmetric NFAs in this work has been done well, novelty seems lack in terms of the chemical structures and the idea of an asymmetric strategy. Also, the manuscript contains many flaws. It is recommended that the authors need to address several issues listed below for the improvement of the manuscript quality.

Response: Thanks for the comments. Asymmetric molecular design of NFAs has been intensively studied and demonstrated effective to improve the device performance of OPVs. However, the asymmetric molecular design in previous typically induces the change of other optoelectronic properties, such as the energy levels, morphological features, which also strongly correlated with the device performance variation. As a result, although high performance has been reached with asymmetric molecular design, it is actually not clear whether does it contribute to the photon-to-electron conversion process or what is contribution from the asymmetric molecular geometry. Therefore, the unambiguous working mechanism and uniqueness of the asymmetric molecular design in OPVs is still not clear. By taking advantages of the rapid progress in “Y” shape NFAs, we design six pairs of symmetric-asymmetric NFA molecules (most of them have not been synthesized and reported in previous work), which show identical optoelectronic properties and morphological feature, to independently study the unambiguous role of molecular asymmetry on device performance. As a result, the major novelty in this work lies in our appropriate design of these twelve molecules, that sufficiently allow us to exclusively unveil the molecular mechanism of asymmetric molecular geometry on the device performance of OPVs. Besides, our findings on the D:A molecular interaction structures via molecular dynamic simulation, and their association with the non-radiative charge recombination and triplet dynamics provide a novel perspective to understand the working mechanism of OPV and should inspire a future avenue of tailoring the D:A

interfacial structure for high-performance OPVs.

Comments:

1. Why are the HOMO and LUMO energy levels of 2BO-EH-based NFAs (2BO-EH-SCI and 2BO-EH-ACI) significantly downshifted compared to 2BO-based NFAs (2BO-SCI and 2BO-ACI), despite having a similar conjugated backbone? Similarly, why did 2BO-based NFAs-based devices exhibit lower efficiencies compared to 2BO-EH-based NFAs-based devices?

Response: We thank the reviewer for the comments. The obvious downshifts of the HOMO/LUMO energy levels and efficiencies are attributed to varied π - π stacking morphology with reduced side chain length, i.e. the 2BO-ACI and BO-EH-ACI (see corrected side chain structures in new manuscript).

The side chain, especially alkyl side chains, are the most widely used solubilizing groups for organic photovoltaics, because they can not only increase the total interaction energy between organic molecules and solvent through the addition of van der Waals interactions, but may also disrupt the molecular or aggregate organization in the solid state (see Ref.: *Chem. Mater.* **26**, 594 – 603(2014)). Considering the limitations of relevant data, we have transformed the objects of two questions into BO-EH-SCI based and C11-BO-SBr based NFAs for further explaining the effects of molecular stacking on energy levels change. As **Supplementary Fig. 29** and **Fig.1d** shown, the GIWAXS results showed that the BO-EH-S/ACI based pristine films exhibited larger d-spacing (3.67 nm) than the C11-BO-S/ABr based pristine films (3.58 nm), indicating higher π - π stacking overlaps may exist in C11-BO-S/ABr based NFAs along the molecular backbone compared to BO-EH-S/ACI based NFAs, which will lead to a stronger electronic coupling (see Ref.: *Nat. Energy* **6**, 605–613 (2021); *Sci. China Chem.* **64**,143 – 1436(2021)), and thus a red-shifted absorption as **Fig. 1b** and narrowed energy gap as **Fig. 1c**. To be specific, shortening the inner alkyl chain of Y6 derivatives, like 2BO-ACI and BO-EH-ACI, will cause the downshifted HOMO and LUMO energy level as reported (see Ref.: *Natl. Sci. Rev.* **7**, 1239–1246(2020)) despite the similar conjugated backbone. The variation of side chains on NFA molecules can induce different crystal packing motifs, thus altering the absorption spectrum, charge transport properties (see Ref.: *Adv. Electron. Mater.* **5**, 1900344(2019)). As reported, decreasing the side chain length increased the lateral distances and reduced the vertical distances of the molecular packing motif for Y6 derivatives with all branched side-chains, resulting in enhanced packing coefficients (see Ref.: *Sci. China Chem.* **64**,143 – 1436(2021)). Hence, the more condensed molecular packing of BO-EH-based NFAs than 2BO-based NFAs helped to build an optimized multi-length-scale morphology, in which the charge dynamic properties were significantly improved, leading to the higher PCE. For example, BO-EH-ACI-based devices ($5.08 \times 10^{-3} / 5.25 \times 10^{-3} \text{ cm}^2 \text{ V}^{-1} \text{ s}^{-1}$) demonstrated apparently higher hole/electron mobility than 2BO-ACI-based devices ($3.58 \times 10^{-3} / 4.14 \times 10^{-3} \text{ cm}^2 \text{ V}^{-1} \text{ s}^{-1}$) as **Supplementary Table 4** shown, which may give rise to lower J_{sc} and FF values, and thus higher PCE of 2BO-based NFAs-based devices than 2BO-EH-based NFAs-based devices. To further express the impact of side chains on PCE, some sentences were added as below. (see **Page 9–10** in the manuscript)

Page 9–10: The NFAs with a similar conjugated backbone showed different energy levels may result from diverse π - π stacking interactions via distinct side-chains, which can induce dissimilar

electronic coupling and energy gap.²¹ Meanwhile, these NFAs with similar backbones but dissimilar side chains will display unequal morphologies, which might be profitable to generate different charge dynamics characteristics and thus PCEs.²⁰

2. Supplementary Fig. 35 shows that there were negligible differences in photoluminescence (PL) quenching in both asymmetric and symmetric binary blends. But, the authors claimed that the asymmetric binary blends exhibited a higher level of PL quenching compared to the symmetric binary blends. It is not understandable. Please provide appropriate comments on this or revise it.

Response: We thank the reviewer for this comment. Since the PL quenching efficiencies gaps between asymmetric and symmetric binary blends were tiny ($< 1\%$), it's hard to figure out the PL intensities differences after the asymmetric and symmetric NFAs blended with PM6. Nevertheless, the pattern on PL quenching efficiencies for asymmetric and symmetric binary blends may contribute to the J_{sc} variation. Therefore, we added the PL quenching intensities as below. (see **Supplementary Fig. 35**). Meanwhile, we revised some sentences as below. (see **Page 18** in the manuscript).

Supplementary Fig. 35 PL spectra of acceptors pure films and PM6: acceptors blend films.

Page 18: 'While compared to the asymmetric binary blends, all the symmetric binary blends showed slightly higher PL quenching efficiency, indicating more effective charge transfer from NFAs to PM6...'

3. Despite an increase in V_{oc} , straight side chain appended asymmetric NFAs-based OSCs exhibited lower efficiencies compared to their symmetric counterparts. What is the limiting factor?

Response: We thank the reviewer for this comment. Besides of the charge recombination, the charge generation properties also have a great impact on the PCE variations. Despite the increase in V_{oc} , the magnitude of decreases in J_{sc} for asymmetric NFAs-based devices is large, resulting in lower PCEs for straight side chains attached asymmetric NFA-based OPVs compared with their corresponding symmetric NFAs-based devices. To further discuss the variation of PCE on symmetric and asymmetric NFAs with straight side-chains, some sentences were added as below.

(see Page 13 in the manuscript)

Page 13: ‘The symmetric NFAs with straight side chains showed significantly better charge generation properties than that of asymmetric NFAs. Despite the increase in V_{oc} , the magnitude of decreases in J_{sc} for asymmetric NFAs-based devices is large, resulting in lower PCEs for asymmetric NFAs with straight side chains compared with their corresponding symmetric NFAs.’

4. Why were C11-BO-SBr and C11-BO-ABr, along with their blends, chosen for GIWAXS analysis over higher-performing blends (PM6:BO-EH-SCl and PM6:BO-EH-ACl)? The GIWAXS data related to PM6:BO-EH-SCl and PM6:BO-EH-ACl should be included in the manuscript.

Response: We thank the reviewer for this comment. The C11-BO-SBr and C11-BO-ABr were chosen for study, for their device performance were first optimized. Afterward, to confirm the generality of our conclusion, we further designed the BO-EH-SCl and BO-EH-ACl, which exhibit similar trend with that of the Br-based systems. Since the GIWAXS measurement is quite dependent on the beam time from synchronic beam lines, we cannot measure all of the systems at our wishes. We totally agree with the reviewer that the GIWAXS of better-performing systems, i.e. the BO-EH-SCl and BO-EH-ACl based symmetric-asymmetric pairs should be examined. Therefore, with the help of Prof. Wei Ma, we sent our samples to the Lawrence Berkeley National Laboratory under Contract No. DE-AC02-05CH11231 with the U.S. Department of Energy, and now we provided the GIWAXS data related to PM6:BO-EH-SCl and PM6:BO-EH-ACl as below (see Fig. 1d and 1e). The detailed description was also provided (see Page 10 in the manuscript). It is clear that the crystallinity variation trend between the symmetric and asymmetric of the BO-EH-SCl and BO-EH-ACl based samples is the same with the BO-EH-SCl and BO-EH-ACl based samples. For example, the pristine acceptor films of BO-EH-SCl and BO-EH-ACl showed the identical packing space and the PM6:BO-EH-SCl and PM6:BO-EH-ACl blend films also showed the crystal coherence length (CCL) values with negligible differences.

Fig. 1 d 2D GIWAXS images of pristine acceptor films and blend films. e GIWAXS intensity profiles of the corresponding films along the in-plane and out-of-plane directions.

Page 10: ‘The pristine acceptor films of BO-EH-SCl and BO-EH-ACl also showed the identical

packing space as **Fig. 1d** and **1e**. Furthermore, the GIWAXS patterns of PM6:BO-EH-SCI (CCL: 23.25 nm) and PM6:BO-EH-ACI (CCL:24.06 nm) blend films also showed little differences with crystal coherence length (CCL), and both blends adopted the face-on molecular orientations. ’

5. The reference 14 in the reference section is incorrect, please check and correct it. It should be ‘Chem. Sci., 2021, 12, 14083–14097’

Response: We thank the reviewer for pointing this out. We apologize for this mistake, and now had revised the reference as ‘Chem. Sci., 2021, 12, 14083–14097’ as below (see **Page 31–32** in the revised manuscript, marked in red).

Page 31 – 32: ‘Gopikrishna, P. *et al.* Impact of symmetry-breaking of non-fullerene acceptors for efficient and stable organic solar cells. *Chem. Sci.* **12**, 14083–14097 (2021).’

6. The HOMO and LUMO labeling in Supplementary Fig. 1 is incorrect, please correct them.

Response: We are grateful to reviewers for the careful reading and pointing this out. As suggested, we had revised the **Supplementary Fig. 1** as below. (see Supplementary Information)

Supplementary Fig. 1 DFT calculation results of molecular orbitals of the Y-shape acceptors with chlorination of end groups on different positions at the B3LYP/6-31G(d) level.

7. The LUMO value of ACI is missing in Supplementary Fig. 2, include it.

Response: We thank the reviewer for this comment. As suggested, we had revised the **Supplementary Fig. 2** as below. (see Supplementary Information)

Supplementary Fig. 2 The HOMO and LUMO energy level of the Y-shape acceptors with simplified side chains as methyl groups-based on the DFT calculation at the long-range corrected ω B97XD/6-31G (d, p) level (e. g. SCI representing C11-BO-SCI, 2BO-SCI or BO-EH-SCI).

8. In line 206 on page 10 in the manuscript, it is written as 'symmetric and symmetric pairs can be derived...' and should be corrected to 'symmetric and asymmetric pairs can be derived...'

Response: We thank the reviewer for this comment. As suggested, we had corrected this typo (see page 10 in the manuscript).

9. In the Figure captions, please write Fig. 30 as Supplementary Fig. 30 and Fig. 31 as Supplementary Fig. 31..

Response: We thank the reviewer for this comment. As suggested, we had corrected the two the Figure captions as **Supplementary Fig. 30** and **Supplementary Fig. 31** (see Supplementary Information).

10. The authors did not mention Supplementary Fig. 33, 37 and 39 in the main text of the manuscript.

Response: We thank the reviewer for this comment. Since none effective information involved in those figures, we previously provided the three figures just for reference. In order to avoid misunderstandings, we removed these figures from the Supplementary Information.

11. The contact angles of the NFAs do not match between Supplementary Fig. 30 and Supplementary Table 2, except for PM6.

Response: We thank the reviewer for this comment. We made a mistake about the position of angle data in **Supplementary Fig. 30**, and we had corrected it as below in **Supplementary Fig. 30**, which match with the **Supplementary Table 3** (see Supplementary Information).

Supplementary Fig. 30 Contact angle images of PM6 and acceptors in thin films with water and diiodomethane droplet on top.

12. The synthesis of C11-BO-SCI, 2BO-SCI and BO-EH-SCI in the supporting information is written as ‘2-(5 or 6-dichloro-3-oxo-2,3-dihydro-1H-inden-1-ylidene)malononitrile (61 mg, 0.27 mmol)’ and should be corrected by removing ‘di’ to ‘2-(5 or 6-chloro-3-oxo-2,3-dihydro-1H-inden-1-ylidene)malononitrile (61 mg, 0.27 mmol)’.

Response: We thank the reviewer for this comment. As suggested, we had corrected the name of ‘2-(5 or 6-dichloro-3-oxo-2,3-dihydro-1H-inden-1-ylidene)malononitrile’ as ‘2-(5 or 6-chloro-3-oxo-2,3-dihydro-1H-inden-1-ylidene)malononitrile’ in the synthetic route of C11-BO-SCI, 2BO-SCI and BO-EH-SCI (see Supplementary Information).

Reviewer #2 (Remarks to the Author):

In this work present by Jinfeng Huang, Tianyi Chen, Le Mei et al. the role of asymmetric molecular geometry on the performance of organic photovoltaics (OSCs) was reported. By designing and synthesizing six pairs of symmetric and asymmetric non-fullerene acceptors with identical physical and optoelectronic properties, the authors found that asymmetric acceptors exhibit higher open-circuit voltages and lower non-radiative charge recombination compared to symmetric counterparts. Molecular dynamics simulations revealed asymmetric acceptors have more diverse donor-acceptor orientations and higher interfacial charge transfer state energy, reducing non-radiative decay. The asymmetric acceptor BO-EH-ACI-based device achieved a high efficiency of 18.80%. Authors highlight the uniqueness of molecular asymmetry for high-performance OSCs by manipulating donor-acceptor interfaces to balance carrier dynamics.

Overall, we feel that this work is not suitable for publication in Nature Communications for the following reasons.

1. Reading through the manuscript, it is so hard to understand what scientifically new findings or new conclusions are claimed in the work. Basically, it is already well known that asymmetric NFAs with different EGs on each side of the molecule exhibit lower charge recombination probability resulting from the higher triplet state energy. So many papers have made such claims using the

methods of electroluminescence-EQE and transient absorption. The same ideas have been reported many times since the first ITIC-type NFA.

Response: We thank the reviewer for this comment. In this work, we claim the asymmetric molecular structure delivers lower non-radiative charge recombination. However, we have never claimed that ‘lower charge recombination probability resulting from the higher triplet state energy’, and this had not ever appeared in our manuscript. In our work, the triplet state energy is not “higher” for asymmetric NFAs based systems. We observed that the triplet intensity in the symmetric NFA based systems becomes higher. While, the higher triplet state formation is not the reason contributes to the lower non-radiative charge recombination, but the result of lower charge recombination probability caused by the higher CT energy, which block the charge back transfer (see Ref.: *Nature* **597**, 666–671 (2021)). We claim that the lower non-radiative charge recombination is due to the higher CT state energy, which is a result of detailed molecular interactions at the D:A interfaces. In this work, we chose to further study the impact of molecular asymmetric geometry on charge recombination, and believe that this study delivers significant novelty due to the following reasons. On the one hand, the accuracy of conclusions on structure-property relationships in OPVs will be affected by multiple variables, including energetic structures, molecular packing, morphology, etc. While the previous study on effect of molecular geometry on device performance of OPV is always entangled with the change of energy levels, morphological factor variations, etc. Therefore, it’s challenging to appropriately design the acceptor molecules that adequately allow us to exclusively unveil the molecular mechanism of asymmetric molecular geometry on the device performance of OPVs. For example, our groups previously reported that more diverse D:A interfacial conformations formed by asymmetric acceptor induced optimized blend interfacial energetics (higher ΔE_{CT-LE}), which contributed to the improved device performance. However, the asymmetric acceptor showed obviously different molecular structures, absorption spectra and energy levels with the symmetric acceptors, (see Ref.: *Nat. Commun.* **13**, 2598 (2022)) so it’s hard to tell if the molecular geometric changes solely influenced the final device performance or due to other reasons. Hence, we carefully designed six pairs of symmetric-asymmetric NFAs with similar photoelectric properties via DFT calculation and experiments to independently study the independent role of molecular asymmetry on the device performance of OPVs.

On the other hand, although many studies have studied the relationship between molecular asymmetry and charge recombination, the underlying influential mechanism is still opaque. To be more specific, understanding the influence of D/A interfaces relevant to charge recombination is lacked in OPV field, thereby preventing the complete realization of the optimum performance potential in simple binary systems. Our work confirmed that the asymmetric NFAs achieved improved efficiency principally by forming a more preferred D/A interface through greater electrostatic potential disparity and dipole moments (see Page 24 - 25 of the revised manuscript), which resulted in a higher CT energy influencing charge generation and recombination. Therefore, if the end groups of NFAs have a strong molecular interaction with donor in the D/A interface with high CT state, it will be more conducive to obtaining high V_{oc} , thereby achieving higher efficiency. As a result, we utilized the commonly concerned asymmetric strategies, but shed a light on the impact of manipulating D/A interfaces on the high device performance, which pave a new way to design more efficient active layers materials and enhance the efficiency of OPVs.

In conclusion, the major novelty in this work lies in our appropriate design of the twelve NFA

molecules, which sufficiently allow us to exclusively reveal the molecular mechanism of asymmetric molecular geometry on the OPV device performance. Moreover, our results on the D:A molecular interaction structures derived from simulation, and their association with non-radiative charge recombination and triplet dynamics, provide a new approach to understand the working mechanism of asymmetric molecular design in OPV, and should inspire a future avenue of tailoring the D:A interfacial structure for high-performance OPVs. For instance, according to the AA-MD simulations, it is a feasible pathway to enhance the V_{oc} by engineering an advantageous D/A interface with high CT state via stronger molecular interactions. To further illustrate the significance of our research, some sentences were added as below (see Page 28 in the manuscript).

Page 28: ‘Our work has revealed the impact of D:A molecular interaction structures on the CT energies for the asymmetric NFAs via AA-MD simulations, and their association with the non-radiative charge recombination and triplet dynamics, which provides a novel perspective to understand the working mechanism of OPV and should inspire a future avenue of tailoring the D:A interfacial structure for high-performance OPVs.’

2. The high performance of 18.80% PCE proposed in this study is not something new either for OSC devices. 18+% PCE has been reported since the beginning of 2022. Fabricating OSC devices with higher performance is good, but essentially it is an engineering problem, or to be specific, nowadays it is a fill factor engineering problem. The scientific part of such research study should focus on the fundamentals behind such high performance. The discoveries proposed in the study should allow readership in the community to develop similar or even higher-performance devices. Repeating the similar performance of devices by testing different combinations of similar D/A materials (or end group combinations) does not advance this field.

Response: We thank the reviewer for this comment. We agreed that 18+% PCE has been reported since the beginning of 2022. However, in our work, we exactly paid more attention to the fundamentals behind such high performance. Our results on the D:A molecular interaction structures derived from simulation, and their association with non-radiative charge recombination and triplet dynamics, provide a new approach to understand the working mechanism of high-performance OPV. Beside, it is worthy of note that most of these efficient devices used the multi-component strategies (see Ref.: *Angew. Chem. Int. Ed.* **63**, 202316039(2024) and *Adv. Mater.* **34**, 2206269(2022)), introduced special treatment on the device fabrication, like the solid additive (see Ref.: *Adv. Energy Mater.* **13**, 2302063(2023), *Adv. Energy Mater.* **13**, 2300763(2023) and *Adv. Mater.* **35**, 2301583(2023)), adopted the layer-by-layer processing (see Ref.: *Adv. Mater.* **35**, 2208279 (2023) and *Nat. Commun.* **14**, 6964 (2023)) or designed the tandem-junction cell (see Ref.: *Adv. Mater.* **34**, 2108090 (2022) and *Joule* **6**, 171–184 (2022)). The foundation of the higher performance based on the above strategies is that the we have good pristine binary active layer materials. Now, we have summarized most of cutting-edge pristine binary OPV, and listed the PCE values as below (summarized in the **Supplementary Table 1**). From this table, it is clear our device performance of 18.80% PCE is still impressive in the pristine binary OPVs.

Supplementary Table 1 Comparison of efficiency and V_{oc} for binary OPVs without special treatment between this work and references.

Active layer	V_{oc} [eV]	PCE[%]	
PM6:BO-EH-ACI	0.927	18.80	This work
PM6:AC9	0.871	18.43	6
PM6:L8-BO	0.87	18.32	7
PM6:L8-HD	0.88	17.32	7
PM6:BP4T-4F	0.839	17.1	8
SZ5:BPT-4F	0.853	16.5	9
SZ5:BPS-4F	0.822	16.1	9
PM6:BTP-S1	0.93	15.21	10
PM6:BTP-S2	0.95	16.37	10
PM6:Y11	0.833	16.54	11
PBDB-T: OY3	0.84	14.51	12
PM6:BO-4Cl	0.841	17.43	13
PM6:CH4	0.888	16.49	14
PM6:CH6	0.875	18.33	14
PBDB-T:DOC2C6-2F	0.85	13.24	15
PTQ10:Y6	0.87	16.21	16
PM6:SY1	0.871	16.83	17
PM6:SY2	0.852	16.01	17
PM6:SY3	0.858	16.23	17
PBT1-C:IDTT-C8-TIC	0.88	13.4	18
PM6:IQx-1	0.911	17.9	19
PM6:BTP-2F-ThCl	0.869	17.06	20
PM6:mBzS-4F	0.804	17.02	21

We agreed that the fill factor (FF) engineering is important in the OPV field. However, according to the well-established empirical relationship below shows that the FF is a function of the V_{oc} , (see Ref.: *Solid State Electron.* **24**, 788 (1981)) indicating that smaller voltage losses (hence larger V_{oc}) also mean larger possible FF for a given material.

$$FF = \frac{\gamma_{oc} - (\gamma_{oc} + 0.72)}{\gamma_{oc} + 1}$$

Where $\gamma_{oc} = qV_{oc}/nkT$, q is the elementary charge, n is the diode ideality factor, k is the Boltzmann constant and T is the temperature. For example, in our work, the BO-EH-based NFAs achieved higher V_{oc} as well as higher FF than the C11-BO-based NFAs, leading to the significantly increased PCEs. Under such circumstances, it's more important to pay attention to the voltage issues in OPV field. Furthermore, the state-of-the-art OPVs usually suffer from high ΔE_{loss} in the range of 0.6–1.1 eV (see Ref.: *Nat. Photon.* **12**, 131–142 (2018)), which is much higher than the theoretical value of 0.25–0.30 eV predicted by Shockley–Queisser (SQ) theory (see Ref.: *Adv. Energy Mater.* **2**, 1100–1108 (2012)). This is because of the relatively high charge recombination loss, especially the strong non-radiative recombination loss. (see Ref.: *Nat. Energy* **1**, 16089 (2016).; *Nat. Mater.* **17**, 703–709 (2018)). A high efficiency of 19% is within reach at wavelengths around 860 ± 60 nm, if the non-radiative recombination V_{oc} loss could be decreased to ~ 0.21 V (see Ref.: *Nat. Mater.* **17**, 119–128 (2018)). Therefore, from the material perspective, effective strategies for decreasing

the non-radiative recombination and understanding the underlying impact mechanisms are key to further enhancing the PCEs of OPVs.

We agree that the research study should focus on the fundamentals behind high performance. However, unveiling the microscopic mechanism underlying the asymmetric molecular design is exactly the significance of our work. Even though some studies have demonstrated the relationship between asymmetry and T1 inhibition, as well as charge recombination, but there are few researches on the underlying mechanisms. By leveraging recent advancements in “Y” shape NFAs, we design six pairs of symmetric-asymmetric NFA molecules (most of them have not been synthesized and reported in previous work) to independently study the unambiguous role of molecular asymmetry on device performance. As a result, the major novelty in this work lies in our appropriate design of these twelve molecules, that sufficiently allow us to exclusively unveil the molecular mechanism of asymmetric molecular geometry on the device performance of OPVs. Besides, our findings on the D:A molecular interaction structures via molecular dynamic simulation, and their association with the non-radiative charge recombination and triplet dynamics provide a novel perspective to understand the working mechanism of OPV and should inspire a future avenue of tailoring the D:A interfacial structure for high-performance OPVs. For example, our work has verified that asymmetric NFAs displaying higher V_{oc} is mainly due to form a more favorable D/A interface via larger electrostatic potential gap and dipole moment, and thus achieved higher CT state energy, subsequently affecting charge generation and recombination. Therefore, it's a feasible pathway to enhance the V_{oc} by engineering the D/A interfaces with strong molecular interactions and high CT states, leading to higher efficiency.

3. In the work, the authors use DFT-based molecular dynamic simulation to study the interactions between the donors and NFAs. The authors proposed that asymmetric NFA exhibits more diverse interactions with the donor phase with a D0-A2 orientation as dominant. However, there are no experimental data to support this statement. The GIWAXS data listing in the supporting information is the same for the symmetric and asymmetric blends. There is no significant diffraction signal that can prove that the claimed orientation only exists in asymmetric NFA blends. And there is no experimental evidence to suggest that there are more diverse D-A interactions. In taking a close look at the computational results solely, it can be seen that all the alkyl chains from both the donor and acceptor were either removed or simplified to the methyl group in the simulation. But in the main text of the manuscript, the authors claimed that they conducted an ‘All-Atoms Molecular Dynamics Simulation (AA-MD)’. So, what the authors mean here for ‘all atoms’ is clearly incorrect. More importantly, it is already so well known that the alkyl chains of the materials play an important role in determining the final morphology of -A BHJ blends. Many high-performance materials that were developed-based on Y6 originate solely from the manipulation of the alkyl chains of Y6, for example, the BTP-EC9 and L8-BO. Thus, it is unacceptable to study the donor-acceptor interaction in simulation without considering the influence of the alkyl chains and thus trying to derive key conclusions of the manuscript-based on such unsupported simulation results.

Response: Thanks for your valuable comment. The real experiment to characterize the molecular packing structures at the D:A interfaces in NFA based bulk-heterojunction blend cannot be implemented based on current science and technology. It is worth noting that the D: A orientations we mentioned were more inclined towards the structural composites of the D:A interfacial

interactions. To avoid misunderstandings, we have modified the relevant concept descriptions on orientations to clearer expressions, such as ‘interfacial structure’ and ‘molecular interaction structure’ (which have been marked in red color in the revised manuscript). Even though the directions and structures of molecular stacking are most likely obtained from single crystal data, it’s almost impossible to cultivate the single crystals with PM6 and NFAs blends for related analysis. The GIWAXS technology, commonly used to investigate OPV blend film morphologies, presents morphological characteristics typically in the scale with tens of nanometers (see Ref.: *Adv. Energy Mater.* **8**, 1–34 (2018); *Nat. Commun.* **7**, 13651 (2016)). However, detailed molecular orientations and overlapping structures at the D:A interfaces are also difficult to be investigated with GIWAXS in symmetric and asymmetric NFA systems. In contrast, all-atom molecular dynamics simulations (AA-MD), a computer-based simulation technique, has been widely used to investigate the relationship of structure at the molecular level (see Ref.: *Chem. Mater.* **29**, 346–354 (2017); *J. Am. Chem. Soc.* **137**, 6254–6262 (2015); *Nature* **515**, 384–388 (2014)). Moreover, previous studies can confirm that our simulation and experiments are in good agreement. For instance, organic semiconductor devices only require local polymer ordering rather than long-range crystallinity, and the computational findings are compared with experimental outcomes from three representative conjugated polymers (see Ref.: *J. Am. Chem. Soc.* **137**, 6254–6262 (2015)). Another example is that the GIWAXS results demonstrated that highly ordered packing of Y6 in BHJ thin films could be formed with the help of solvent additive 1-chloronaphthalene (CN). Our AA-MD calculation shows that CN can act as a bridge for Y6 molecules to connect and stack with each other to form long-range ordering in thin films (see Ref.: *J. Mater. Chem. A*, **11**, 21895–21907 (2023)). Based on the aforementioned aspects, we utilize the AA-MD tools to compensate for the experiment.

Despite unable to gather accurate data on D: A interfacial structures from the experiment, two different perspectives through electrostatic potential (ESP) and dipole moment were further introduced for determining the D:A molecular interaction structures and the outcomes matched with the MD results (see new manuscript). As shown in **Fig. 4a**, the D_0 fragments of donor PM6 display a negative ESP value on most of the molecular surface, indicating its electron-rich nature. In contrast, the ESP values are positive on most of the IC-XBr ($X = 0, 1, 2$) and A_0 surfaces, suggesting an electronic affinity. The ESP values of every atoms on the five fragments were further given in **Supplementary Table 5**. The atoms on D_0 moiety showed obviously average negative ESP values compared to A_0 moiety, indicating higher ESP gap with the IC-XBr ($X = 0, 1, 2$) fragments with positive ESP values in the D/A interfaces. The increased ESP difference between donor and acceptor will enhance the intermolecular interaction, suggesting stronger intermolecular interaction for the D_0 and IC-XBr ($X = 0, 1, 2$) fragments compared to the counterpart of A_0 and IC-XBr ($X = 0, 1, 2$) fragments. With more bromine atoms, most atoms on IC-XBr ($X = 0, 1, 2$) showed higher ESP values because of the electron-withdrawing properties of the bromine atom. However, the bromine atoms at bottom of the IC-XBr ($X = 0, 1, 2$) labeled as 18 and 19 (see **Fig.4 c** and **Supplementary Table 5**) showed significantly decreased ESP value, which may attribute to the larger electronegativity (χ) value of bromine (2.8) than hydrogen (2.1) and carbon (2.5) atoms (see Ref.: *J. Am. Chem. Soc.* **112**, 4741–4747 (1990)). It’s worth noting that the 18th and 19th atoms on IC displayed the highest ESP values among all the atoms on the IC-XBr ($X = 0, 1, 2$), leading to the highest dipole moment as 4.66 D among the three end groups of NFAs (IC-Br: 4.03 D; IC-2Br: 3.14 D). The larger dipole moment of end group will lead to a stronger intermolecular interaction with the PM6 donor, especially the D_0 moiety. Herein, we calculate the interaction energy of

different complex with different moiety pairs, and we found that D_0-A_2 showed the highest interaction energy, which is consistent with the ESP and dipole moment results as above, leading to a dominant interaction molecular interaction structure.

We did not claim the D_0-A_2 orientation only exists in asymmetric NFA blends, but the interaction energies and contact probabilities of several types of D:A interfacial structures are compared. Since the asymmetric NFA have one more kind of end group than the symmetric NFA, it's reasonable that more diverse D:A interactions will exist in the D/A interfaces and will lead to more energetic disturbance. As shown in **Supplementary Fig. 32**, the calculated Urbach energy (E_U) of asymmetric BO-EH-ACl-based device is 19.5 meV, which is slightly larger than that of the symmetric BO-EH-SCI-based device as 18.8 meV, suggesting that the asymmetric structure may bring more energetic disturbance due to more diverse D:A interactions. Similar observation has been reported by Tobin et al., that voltage loss of the asymmetric acceptors is still lower than that of the symmetric counterparts (see Ref.: *Joule* 7, 2152–2173 (2023)), so the increased ΔV_2 caused by disorder may not affect the overall V_{oc} loss.

We conducted the AA-MD simulations for the BO-EH-SBr and BO-EH-ABr symmetric-asymmetric pair with donor PM6, and in the whole process, the alkyl chains of donor and acceptors were not removed. In **Fig. 4** in the manuscript, the aim for removing all the alkyl chains of donor and acceptors is to observe diverse D-A pairs more clearly, which had already been explained in the Figure captions. To avoid misunderstandings, we revised the related images in **Fig. 4** as below (See **Page 25** in the new manuscript).

Page 25:

Fig. 4b Intermolecular stacking configurations of the complexes with side-chains.

In electronic-structure calculations, since the alkyl chains attached to π -conjugated backbones have little impact on electronic properties of the whole molecules/polymers, it is thus a common practice for computational chemists to substitute long alkyl chains with relatively smaller methyl or hydrogen atom. Moreover, for our TDDFT calculations, we focus on the charge-transfer state in the blended thin film, and the charge will transfer along the molecular π -conjugated backbone rather

than the alkyl chains (see Ref.: *J. Mater. Chem. A*, **9**, 16733-16742 (2021); *Nat. Commun.* **11**, 3943 (2020)).

Reviewer #3 (Remarks to the Author):

In this manuscript, the authors studied the independent role of molecular asymmetry on the device performance of OPVs. They have designed and synthesized six pairs of symmetric-asymmetric non-fullerene acceptors (NFAs), which exhibited identical physical and optoelectronic properties for each symmetric and asymmetric NFA pair. Compared to their symmetric counterparts, these asymmetric NFAs also demonstrated increased open-circuit voltage and lower non-radiative charge recombination loss. With carrier dynamic measurements and MD simulations, the authors unveiled that the asymmetric NFA-based OPV exhibit more diverse D:A interfacial structures with averagely higher charge transfer states, and less T1 formation, which directly contributes to the reduced non-radiative charge recombination. As a result, best device performance is reached with asymmetric NFAs, with power conversion efficiency of 18.8%, and is one of the highest efficiency among binary OPVs. This work illustrate the working mechanism of molecular asymmetry on the device performance of OPVs, provides a new perspective of manipulating the D:A interfacial contact structures, and should have important implications on molecular structure design for high-performance OPVs. Therefore, the reviewer recommends this work to be published in *Nat. Commun.* after addressing the following issues.

1. As mentioned in this paper, both higher CT energy and T1 suppression are conducive to inhibit the non-radiative voltage loss. The link between the asymmetric molecular structure and the T1 formation is weak, and this part should be discussed in more details.

Response: We thank the reviewer for this comment. As shown in **Fig. 3i**, the non-geminate charge recombination from CS state will form the ^3CT and ^1CT state with ratio of 3:1. The ^3CT and ^1CT states are close in energy, since their energy difference is just electronic exchange energy that is positively proportional to overlap of wavefunctions of electron localized at acceptor and hole localized at donor. The back transfer from ^3CT to T1 state is an important non-emissive decay pathway, which will increase the non-radiative voltage loss. The kinetics of non-radiative decay between two molecular electronic states can be described by Marcus' equation derived via Fermi Golden Rule, which suggests that a larger energy gap between ^3CT and T1 suppresses $^3\text{CT}\rightarrow\text{T1}$ rate constant (see Ref.: *Adv. Energy Mater.* **7**, 1602713(2017)). Since the ^3CT state usually has a higher energy level than T1 state, under the assumption that the approximate T1 energy level for two similar NFAs, it is reasonable to understand that the occurrence probability for the $^3\text{CT}\rightarrow\text{T1}$ process is lower in asymmetric structures because of their higher CT energy. To strength the link between the asymmetric molecular structure and the T1 formation, some sentences were added as below (see **Page 20** in the manuscript).

Page 20: The rate constant of non-radiative decay between two molecular electronic states can be described by Marcus' equation derived via Fermi Golden Rule, which suggests that a larger energy gap between ^3CT and T1 suppresses $^3\text{CT}\rightarrow\text{T1}$ rate constant.³¹ The ^3CT state usually has a higher energy level than T1 state. Under the assumption that the approximate T1 energy level for two

similar NFAs, it is reasonable to understand that the higher CT energy of asymmetric NFAs allowed less production of triplet excitons with lower occurrence probability for the $^3\text{CT} \rightarrow \text{T1}$ process.'

2. The D:A interfacial contact structures is very important for optoelectronic properties of OSCs. Besides of the charge recombination, I believe this will also dramatically affect the charge generation process. Moreover, the asymmetric structures tend to form a interfacial dipole moments, and I suggest the authors discuss their effects on the charge generation.

Response: We thank the reviewer for this comment. We agreed that the functionality of D/A interfaces in simultaneously regulating both free charge generation and recombination, which determines the device efficiency. The asymmetric NFAs displayed higher CT energy via more favourable D/A interfaces, which is beneficial for asymmetric NFAs to achieve a lower energetic offset between the Energy bandgap (E_g) and the charge transfer state ($\Delta E_{\text{CT}} = E_g - E_{\text{CT}}$) than the symmetric counterpart. Lower ΔE_{CT} was adverse to efficient charge separation and thus less efficient charge generation, which is consistent with the experimental results.

It's rational to form interfacial dipole moments by the asymmetric structures. However, due to the complicated molecular directions and angles, it's hard to comprehensively characterizing the intermolecular interactions at D/A interfaces merely via dipole moments. Hence, we combined the electrostatic potential (ESP) and dipole moments to discuss their effects on the charge generation as **Fig. 4**. As suggested, we provided the ESP of five fragment structures, including IC (A_1), IC-Br ($A_1=A_2$) and IC-2Br (A_2) for two acceptors (BO-EH-SBr and BO-EH-ABr), and D_0 and A_0 for donor PM6 in **Supplementary Table 5**. Basically, the electrostatic potential (ESP) and dipole moments determined more favorable D/A interfaces and thus E_{CT} , leading to different ΔE_{CT} affecting the charge generation.

To address the effects of ESP and interfacial dipole moments on the charge generation, a paragraph and some sentences were added as below. (see **Page 22–25** in the manuscript)

Page 22 – 23: 'Before conducting the AA-MD, we calculated the Electrostatic potential (ESP) of five relevant fragments in the D:A interfaces as reference. As shown in **Fig. 4a**, the D_0 fragments of donor PM6 display a negative ESP value on most of the molecular surface, indicating its electron-rich nature. In contrast, the ESP values are positive on most of the IC-XBr ($X = 0, 1, 2$) and A_0 of donor PM6 surfaces, suggesting an electronic affinity. The ESP values of every atoms in the five fragments were further given in **Supplementary Table 5**. The atoms in D_0 moiety showed obviously average negative ESP values compared to A_0 moiety, indicating a higher ESP gap of the former with the IC-XBr ($X = 0, 1, 2$) fragments with positive ESP values in the D/A interfaces. The increased ESP difference between donor and acceptor will enhance the intermolecular interaction, suggesting stronger intermolecular interaction for the D_0 and IC-XBr ($X = 0, 1, 2$) fragments compared to the counterpart of A_0 and IC-XBr ($X = 0, 1, 2$) fragments. With more bromine atoms, most atoms on IC-XBr ($X = 0, 1, 2$) showed higher ESP values because of the electron-withdrawing properties of the bromine atom. However, the bromine atoms at bottom of the IC-XBr ($X = 0, 1, 2$) labeled as 18 and 19 showed significantly decreased ESP value (**Fig. 4c**), probably attributed to the larger electronegativity (χ) value of bromine (2.8) than hydrogen (2.1) and carbon (2.5) atoms.³⁵ It's worth noting that the 18th and 19th atoms on IC displayed the highest ESP values among all the atoms on the IC-XBr ($X = 0, 1, 2$), leading to the highest dipole moment as 4.66 D among the three

end groups of NFAs (IC-Br: 4.03 D; IC-2Br: 3.14 D). The larger dipole moment of end group will lead to a stronger intermolecular interaction with the PM6 donor, especially the D_0 moiety.’

Page 24: ‘Adjusting the fragments for PM6 with the same atomic numbers, the optimal molecular orientation was maintained as **Supplementary Fig. 38.**’

Page 25: ‘Meanwhile, it is also conducive for asymmetric NFAs to achieve a lower energetic offset between the Energy bandgap (E_g) and the charge transfer state ($\Delta E_{CT} = E_g - E_{CT}$) than the symmetric counterpart. Lower ΔE_{CT} was adverse to efficient charge separation and thus less efficient charge generation, which is consistent with the experimental results.’

Fig. 4 The electrostatic potential (ESP), Molecular dynamics simulations and TDDFT calculations for PM6:BO-EH-SBr and PM6:BO-EH-ABr complexes. **a** The ESP distribution images of five fragment structures. **b** Intermolecular stacking configurations of the complexes with side-chains. **c** ESP values of two atoms on three fragments. **d** Interaction energy, and **e** contact probability. **f-h**

The distribution of CT energy of donor PM6 with end groups (A1, A2) of different acceptors.

Supplementary Table 5 Average ESP of each atom of five fragments calculated on B3LYP/6-31G(d, p) level.

Atom Number	ESP(IC) [kJ mol ⁻¹]	ESP(IC-Br) [kJ mol ⁻¹]	ESP(IC-2Br) [kJ mol ⁻¹]	ESP(A ₀) [kJ mol ⁻¹]	ESP(D ₀) [kJ mol ⁻¹]	Atom Number	ESP(D ₀) [kJ mol ⁻¹]
1	11.47	13.94	15.53	0.51	-5.11	23	6.51
2	15.26	18.07	19.60	2.02	-4.66	24	4.63
3	11.93	15.36	17.54	2.00	-2.78	25	5.63
4	10.33	13.71	15.99	0.56	-5.15	26	8.47
5	13.17	15.71	17.70	6.14	-4.71	27	-8.00
6	6.72	10.19	12.33	4.63	-2.64	28	-8.59
7	7.04	10.20	11.50	2.04	-5.90	29	10.68
8	6.67	9.05	11.14	2.00	-5.94	30	11.74
9	7.20	10.46	12.83	4.56	-4.80	31	11.50
10	-15.48	-13.35	-11.87	0.52	-5.75	32	11.03
11	12.78	15.35	16.67	6.14	-6.48		
12	6.65	8.52	9.64	0.58	-6.48		
13	-17.32	-16.07	-15.10	-21.45	-4.49		
14	6.27	8.78	10.05	-21.52	-4.54		
15	-17.90	-16.38	-15.28	10.29	-4.37		
16	12.98	16.18	14.04	10.26	-3.07		
17	10.61	8.48	10.12	10.30	-4.83		
18	19.11	2.84	3.35	10.32	-0.77		
19	20.19	18.53	4.83		-4.54		
20	7.49	9.75	11.22		-2.21		
21	10.12	12.20	13.13		-5.27		
22	9.66	11.67	13.54		-2.35		

3. Although there is a reliable variation trend on the short circuit current density (J_{sc}) and V_{oc} values between symmetric and symmetric pairs as Fig. 2b shown, the PCE variations for acceptors with straight side chains, like C11-BO-SX and C11-BO-AX (X=F, Cl, Br), were totally different from

the NFAs with all branched side chains, like 2BO-SCl and 2BO-ACl, BO-EH-SX and BO-EH-AX (X= Cl, Br). This part should be discussed in more details.

Response: We thank the reviewer for this comment. The charge generation and recombination properties have a great impact on the PCE variations. On the one hand, despite the increase in V_{oc} , the the magnitude of decreases in J_{sc} and FF for asymmetric NFAs-based devices were much larger, resulting in lower PCEs for straight side chains attached asymmetric NFA-based OPVs compared with their corresponding symmetric NFAs-based devices. On the other hand, the improved charge dynamics properties, resulting in improved J_{sc} and FF, coordinated assistance to the higher V_{oc} for obtaining higher efficiencies of all branched side chains attached asymmetric NFAs compared to their symmetric counterparts.

To further discuss the variation of PCE on symmetric and asymmetric NFAs with different side-chains, some sentences were added as below. (see **Page 13** in the manuscript)

Page 13: ‘On the one hand, the symmetric NFAs with straight side chains showed significantly better charge generation properties than those of asymmetric NFAs. Despite the increase in V_{oc} , the magnitude of decreases in J_{sc} for asymmetric NFAs-based devices were much larger, resulting in lower PCEs for asymmetric NFAs with straight side chains compared with their corresponding symmetric NFAs. On the other hand, the improved charge dynamics properties resulted in improved J_{sc} and FF, which coordinated assistance to the higher V_{oc} to obtain higher efficiencies of asymmetric NFAs with all branched side chains compared to the symmetric ones.’

4. The authors studied the light intensity dependent voltage and photo-current density, and the corresponding results are presented in Supplementary Fig. 38 in the charge transfer, transport and recombination part of the supporting information. However, there are no relevant results and discussions provided in the main text.

Response: We thank the reviewer for this comment. Since none effective information involved in **Supplementary Fig. 38**, we previously provided the **Supplementary Fig. 38** just for reference. In order to avoid misunderstandings, we removed it from the Support Information.

5. Figures in this work should be polished. For example, the Fig. 1b contains two Figures with similar information, and the illustrations inside the Figures is not concise.

Response: We thank the reviewer for this comment. We have polished some images in **Fig. 1** and **4** as below (see **Page 7** and **25** in the manuscript).

Page 7:

Fig. 1 Basic characteristics of the donor PM6 and six pairs of symmetric and asymmetric acceptors. **a** Molecular structures of polymer donor PM6 (poly((4,8-bis(5-(2-ethylhexyl)-4-fluoro-2-thienyl)benzo[1,2-b:4,5-b']dithiophene-2,6-diyl)-2,5-thiophenediyl(5,7-bis(2-ethylhexyl)-4,8-dioxo-4H,8H-benzo[1,2-c:4,5-c']dithiophene-1,3-diyl)-2,5-thiophenediyl)) and acceptors (D represents electron rich unit in blue dashed box, and A represents deficient unit in red dashed box and π represents conjugated bridging unit in the gray circle). **b** Normalized absorption spectra of twelve acceptors in CHCl₃ solution and thin films. **c** Energy level alignment of PM6 and all acceptors via CV measurement. **d** 2D GIWAXS images of pristine acceptor films and blend films. **e** GIWAXS intensity profiles of the corresponding films along the in-plane and out-of-plane directions.

Fig. 4 The electrostatic potential (ESP), Molecular dynamics simulations and TDDFT calculations for PM6:BO-EH-SBr and PM6:BO-EH-ABr complexes. **a** The ESP distribution images of five fragment structures. **b** Intermolecular stacking configurations of the complexes with side-chains. **c** ESP values of two atoms on three fragments. **d** Interaction energy, and **e** contact probability. **f-h** The distribution of CT energy of donor PM6 with end groups (A_1 , A_2) of different acceptors.

Even though we had explicated the change characteristics for NFAs with different end groups and side chains in films as Fig. 1b, in terms of their absorption in solution, there were relatively few related descriptions. As suggested, we have added some sentences as below (see **Page 9** in the manuscript).

Page 9: ‘Every pair of symmetric-asymmetric NFAs showed the identical absorption region with maximum peaks as well as onsets. For example, BO-EH-SCl and BO-EH-ACl showed the same peaks and onsets at 727 and 784 nm, respectively, in dilute solution.’

6. Since end group tailoring is the key molecular design strategy in this work, more relevant references should be cited.

Response: We thank the reviewer for this comment. We had added more relevant references as below (see **Page 31–33** in the manuscript).

Page 31 – 32:

‘16. Gopikrishna, P. *et al.* Impact of symmetry-breaking of non-fullerene acceptors for efficient and stable organic solar cells. *Chem. Sci.* **12**, 14083–14097 (2021).’

Page 32:

‘21. Li, C. *et al.* Non-fullerene acceptors with branched side chains and improved molecular packing to exceed 18% efficiency in organic solar cells. *Nat. Energy* **6**, 605–613 (2021).’

Page 33:

‘31. Chen, X. *et al.* Suppressing Energy Loss due to Triplet Exciton Formation in Organic Solar Cells: The Role of Chemical Structures and Molecular Packing. *Adv. Energy Mater.* **7**, 1602713 (2017).

35. Jeffrey K. N. *et al.* Atomic Polarizability and Electronegativity. *J. Am. Chem. Soc.* **112**, 4741-4747 (1990).’

REVIEWERS' COMMENTS

Reviewer #1 (Remarks to the Author):

The authors have made significant efforts in synthesizing NFA molecules and characterizing their photovoltaic properties. The manuscript became more flawless after revision. Nevertheless, novelty is not enough to be published in Nature Communications, and a clear-cut take-home message cannot be found.

Reviewer #2 (Remarks to the Author):

The authors have addressed all of my concerns and the manuscript is ready for publication

Reviewer #3 (Remarks to the Author):

The authors have made a reasonable response and careful revision. I recommend the publication of this manuscript in Nature Communications.